# Deletion of the autism-related gene *Chd8* alters activity-dependent transcriptional responses in mouse postmitotic neurons

Atsuki Kawamura[1] & Masaaki Nishiyama [1 ✉]

*CHD8* encodes chromodomain helicase DNA-binding protein 8 and its mutation is a highly penetrant risk factor for autism spectrum disorder (ASD). CHD8 serves as a key transcriptional regulator on the basis of its chromatin-remodeling activity and thereby controls the proliferation and differentiation of neural progenitor cells. However, the function of CHD8 in postmitotic neurons and the adult brain has remained unclear. Here we show that *Chd8* homozygous deletion in mouse postmitotic neurons results in downregulation of the expression of neuronal genes as well as alters the expression of activity-dependent genes induced by KCl-mediated neuronal depolarization. Furthermore, homozygous ablation of CHD8 in adult mice was associated with attenuation of activity-dependent transcriptional responses in the hippocampus to kainic acid–induced seizures. Our findings implicate CHD8 in transcriptional regulation in postmitotic neurons and the adult brain, and they suggest that disruption of this function might contribute to ASD pathogenesis associated with CHD8 haploinsufficiency.

[1] Department of Histology and Cell Biology, Graduate School of Medical Sciences, Kanazawa University, Kanazawa, Ishikawa 920-8640, Japan.
✉email: nishiyam@staff.kanazawa-u.ac.jp

Autism spectrum disorder (ASD) is a heterogeneous condition defined by deficits in social interaction and communication as well as by restricted and repetitive behaviors. Individuals with ASD often manifest additional symptoms such as seizures, anxiety, and intellectual disability[1]. Synaptic dysfunction is a key feature of ASD pathology and is also apparent in the brain of various mouse models of the disease[2–4]. Neuronal activity triggered by experiences contributes to the regulation of synaptic development and function in neural circuits by inducing the transcription of multiple genes[5,6], suggesting that dysfunction of such activity-dependent transcriptional regulation may contribute to the development of ASD.

Mutations in the gene encoding chromodomain helicase DNA-binding protein 8 (CHD8) constitute a highly penetrant risk factor for ASD[7,8]. CHD8 is an ATP-dependent chromatin-remodeling factor that targets the promoter regions of many genes, including those of other genes associated with ASD, and thereby regulates their transcription[9–11]. Chd8 heterozygous mutant mice have been found to manifest macrocephaly, increased anxiety-like behavior, altered social behavior, and cognitive deficits, but the behavioral phenotypes of different Chd8 mutant mouse lines generated by different groups overlap only partially[12–17]. Loss of CHD8 also results in impaired proliferation and differentiation of precursor cells for excitatory neurons of the forebrain and for cerebellar granule cells during cortical and cerebellar development[18,19]. Furthermore, CHD8 plays a key role in oligodendrocyte differentiation and myelination, and its ablation in oligodendrocyte precursor cells of mice results in the development of some of the behavioral phenotypes characteristic of Chd8 heterozygous mutant mice[20–22]. Although these various observations implicate CHD8 as a central regulator of the proliferation and differentiation of progenitor cells in the brain, whether CHD8 also plays an important role in postmitotic neurons and the adult brain has been unknown.

We have now examined the consequences of Chd8 deletion in mouse postmitotic neurons both in vitro and in the adult brain in vivo with the use of a Cre recombination system inducible by tamoxifen. We found that CHD8 regulates the expression of neuronal genes and activity-dependent genes in cultured neurons. We also found that deletion of Chd8 in the adult brain results in downregulation of activity-dependent gene expression associated with kainic acid (KA)–induced seizures. Our results indicate that CHD8 serves as a transcriptional regulator not only in neural progenitor cells but also in postmitotic neurons.

## Results

**Activity-dependent gene expression is attenuated by Chd8 ablation in postmitotic neurons in vitro.** To investigate the role of CHD8 in postmitotic neurons, we cultured primary hippocampal neurons derived from Cre recombinase–mediated Chd8 knockout (CAG-CreER/Chd8$^{F/F}$) and control (Chd8$^{F/F}$) mice at embryonic day (E) 18.5 (Fig. 1a). The cultures were treated with cytosine β-D-arabinofuranoside (Ara-C) to eliminate dividing cells and were then exposed to 4-hydroxytamoxifen (4-OHT) to generate Chd8 conditional knockout (Chd8 CKO) neurons. The abundance of Chd8 mRNA in Chd8 CKO cultures was greatly reduced compared with that in control cultures (Fig. 1b). The cultured cells comprised ~17% astrocytes and ~1% oligodendrocytes in addition to neurons, but the proportion of each cell type and cell viability were similar for Chd8 CKO and control cultures (Fig. 1c, Supplementary Fig. 1). We performed RNA-sequencing (RNA-seq) analysis with these Chd8 CKO and control neurons (Supplementary Table 1). The expression of 509 genes was downregulated and that of 360 genes was upregulated in Chd8 CKO neurons compared with control neurons (false discovery rate (FDR)–adjusted P value of <0.05) (Fig. 1d). Gene

ontology (GO) analysis revealed that the 509 downregulated genes in Chd8 CKO neurons were enriched in genes related to "translation," "transcription, DNA-templated," "nervous system development," and "positive regulation of synapse assembly," whereas the 360 upregulated genes showed significant enrichment for genes related to "cellular amino acid metabolic process" and "transport" (Fig. 1e). SynGO analysis revealed that the downregulated genes in Chd8 CKO neurons were enriched in genes related to pre- and postsynaptic function, whereas the upregulated genes showed no significant enrichment (Fig. 1f, Supplementary Fig. 2). Furthermore, gene set enrichment analysis (GSEA) for Kyoto Encyclopedia of Genes and Genomes (KEGG) pathways showed that ribosomal genes were significantly downregulated in Chd8 CKO neurons (Fig. 1g).

Neuronal depolarization mediated by an increase in the extracellular KCl concentration induces the expression of activity-dependent genes[23]. We therefore next examined whether the induction of gene expression in response to neuronal activity is altered by Chd8 ablation in cultured postmitotic neurons. Control neurons treated with 55 mM KCl for 2 h showed a significant increase in the expression of activity-dependent genes such as Arc, Egr1, Fos, Fosb, Npas4, and Nr4a1 compared with those treated with 5 mM KCl (Fig. 2a, Supplementary Table 2). The peak mRNA abundance occurred at ~1 h for Egr1 and at ~2 h for Fosb and Nr4a1, with the amounts of these mRNAs declining gradually thereafter (Supplementary Fig. 3a–c). For neurons treated with 55 mM KCl, the expression of 775 genes was downregulated and that of 404 genes was upregulated by Chd8 ablation (FDR-adjusted P < 0.05) (Fig. 2b, Supplementary Table 3). Among these differentially expressed genes, the expression of activity-dependent genes—including Egr1, Fosb, and Nr4a1—was significantly downregulated in Chd8 CKO neurons (Fig. 2b, c). Immunoblot analysis confirmed that the activity-dependent expression of FOSB at the protein level was significantly attenuated in Chd8 CKO neurons compared with control neurons (Supplementary Fig. 3d–f). We also examined our RNA-seq data for the 190 KCl-induced genes with a log$_2$(fold change) of >2.0 associated with an FDR-adjusted P value of <0.01 in control neurons treated with 55 mM KCl compared with those treated with 5 mM KCl (Fig. 2a). GSEA revealed that the expression of these KCl-induced genes was downregulated in Chd8 CKO versus control neurons under the 55 mM KCl condition (Fig. 2d). In addition, SynGO analysis and GSEA for KEGG pathways revealed that genes with significantly downregulated expression in Chd8 CKO neurons versus control neurons under this condition included those related to synapses and ribosomes (Supplementary Fig. 4a–c). Comparison of differentially expressed genes (FDR-adjusted P < 0.05) between Chd8 CKO and control neurons under the 5 and 55 mM KCl conditions showed that 227 upregulated and 426 downregulated genes were specifically identified by 55 mM KCl treatment (Fig. 2e, Supplementary Fig. 4d–f, Supplementary Table 4). GO analysis revealed that these downregulated genes were enriched in genes related to "transcription, DNA-templated," "mRNA processing," "nervous system development," and "cellular response to calcium ion" (Fig. 2f).

**Chromatin accessibility profiling at CHD8 binding sites for neurons activated by KCl-induced depolarization.** Given the role of CHD8 as a chromatin-remodeling factor, we performed assay for transposase-accessible chromatin (ATAC)-seq analysis to assess genome-wide chromatin accessibility in Chd8 CKO and control neurons after treatment with 5 or 55 mM KCl for 2 h. Most of the ATAC-seq peaks (50.2 to 78.2% of total peaks) were shared among these four conditions (Fig. 3a). Heat map and density profiles around high-confidence CHD8 chromatin immunoprecipitation (ChIP)–seq peaks of previously published data[12] revealed

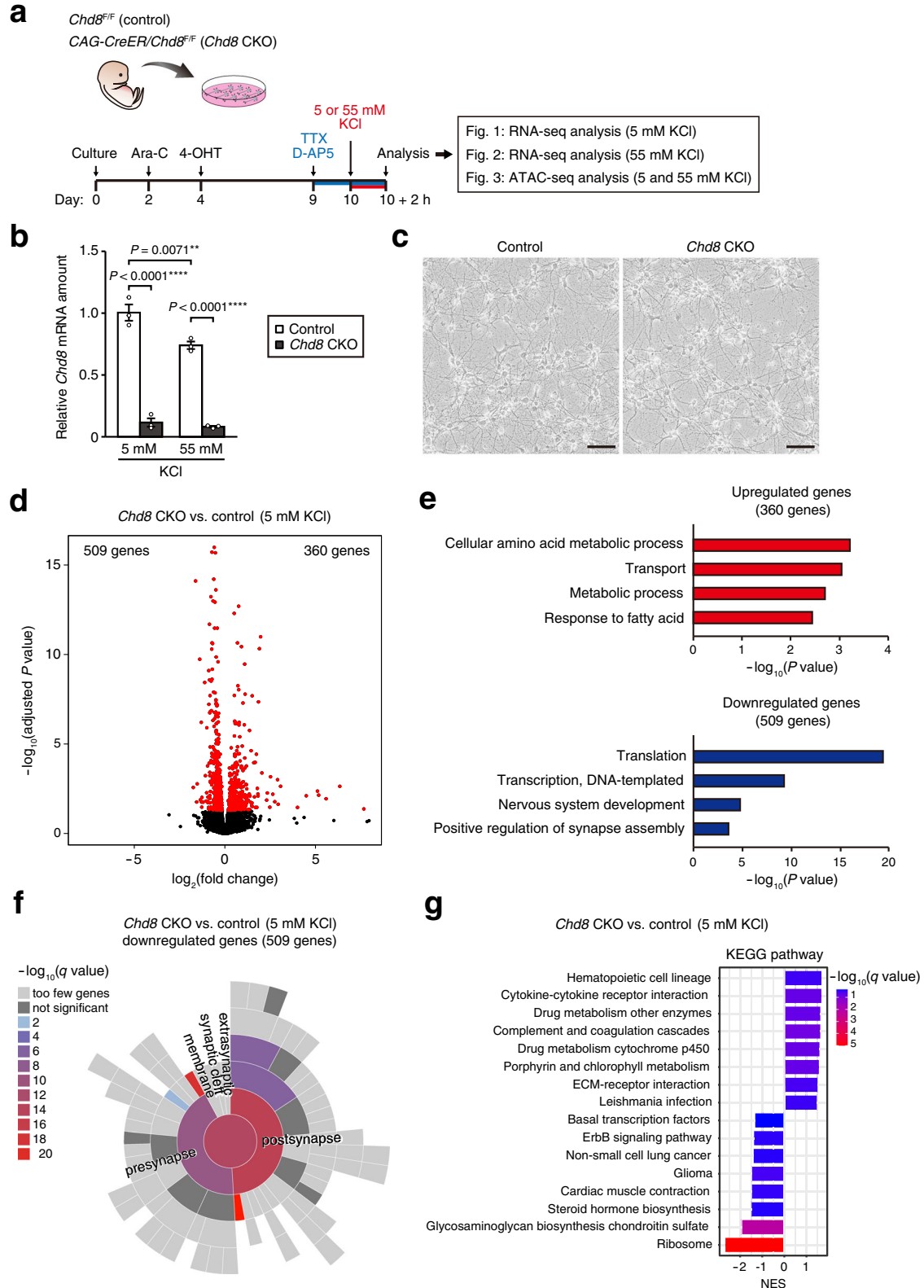

similar distribution patterns of ATAC-seq signal intensity between *Chd8* CKO and control neurons (Fig. 3b, Supplementary Fig. 5a). We also examined chromatin accessibility at loci of several activity-dependent genes including *Fosb*, *Nr4a1*, *Homer1*, and *Egr3*, all of which were downregulated in *Chd8* CKO neurons compared with control neurons under the 55 mM KCl condition. CHD8 peaks that overlapped with accessible chromatin sites were detected at or near

these genomic loci (Fig. 3c). Although the signal intensity of several ATAC-seq peaks in these regions increased in response to neuronal activity, the signal distribution patterns were similar for the two genotypes (Fig. 3c, Supplementary Fig. 5b). In addition, we performed ChIP-quantitative polymerase chain reaction (qPCR) analysis to examine the effects of *Chd8* deletion on trimethylated Lys[4] of histone H3 (H3K4me3) abundance and RNA polymerase II

**Fig. 1 Translation- and neuron-related gene expression is downregulated by *Chd8* ablation in postmitotic neurons. a** Schematic representation of the experimental procedure for culture of primary neurons, deletion of *Chd8* by Cre-mediated recombination, and KCl-induced neuronal depolarization (see Methods for details). The Na$^+$-channel blocker tetrodotoxin (TTX) and *N*-methyl-D-aspartate (NMDA) receptor antagonist D-AP5 were added to neurons in order to reduce neuronal activity before KCl-induced depolarization. **b** Reverse transcription (RT) and real-time polymerase chain reaction (PCR) analysis of *Chd8* mRNA in control and *Chd8* CKO neurons treated with 5 or 55 mM KCl. Data are means ± s.e.m. (*n* = 3 mice of each genotype). **\*\*P* < 0.01, **\*\*\*\*P* < 0.0001 (one-way ANOVA with Tukey's post hoc test). **c** Phase-contrast micrographs of primary neurons cultured for 10 days. Scale bars, 50 μm. **d** Volcano plot for differentially expressed genes in *Chd8* CKO neurons compared with control neurons under the 5 mM KCl condition as determined by RNA-seq analysis (*n* = 3 mice of each genotype, 1 male and 2 females for *Chd8* CKO mice and 2 males and 1 female for control mice). Differentially expressed genes (FDR-adjusted *P* value of <0.05) are highlighted in red. **e** GO analysis of genes whose expression was upregulated (360 genes) or downregulated (509 genes) in *Chd8* CKO neurons. **f** SynGO analysis of genes whose expression was downregulated (509 genes) in *Chd8* CKO neurons. **g** GSEA for KEGG pathways in *Chd8* CKO neurons compared with control neurons under the 5 mM KCl condition. NES, normalized enrichment score.

recruitment. The extent of H3K4me3 or RNA polymerase II enrichment around the transcription start site (TSS) of several activity-dependent genes tended to be higher in *Chd8* CKO neurons than in control neurons under the 55 mM KCl condition, but these differences did not achieve statistical significance (Fig. 3d, e).

**Activity-dependent gene expression is attenuated by *Chd8* ablation in the adult brain in vivo.** We next generated mice with CHD8 deficiency in adulthood (hereafter referred to as *Chd8* CKO mice) by administering tamoxifen to 8- to 12-week-old *CAG-CreER/Chd8*$^{F/F}$ mice intraperitoneally for five consecutive days. We confirmed that the floxed (F) alleles of *Chd8* were efficiently deleted in the hippocampus of *Chd8* CKO mice as reflected by the loss of *Chd8* expression at the mRNA and protein levels (Fig. 4a, Supplementary Fig. 6a, b). To examine whether CHD8 regulates the expression of activity-dependent genes in vivo, we induced widespread neuronal activity in the brain of *Chd8* CKO and control mice by injection of the glutamate receptor agonist KA. Although we did not detect a significant difference in seizure severity between *Chd8* CKO and control mice after KA treatment (Fig. 4b), the expression of activity-dependent genes including *Fosb*, *Nr4a1*, and *Egr1* in the hippocampus was significantly downregulated in *Chd8* CKO mice relative to control mice with a similar seizure stage at 60 min after KA treatment (Fig. 4c). The abundance of FOSB protein was also significantly reduced in the hippocampus of *Chd8* CKO mice compared with that of control mice after KA treatment (Supplementary Fig. 6a, c). To examine whether *Chd8* ablation alters chromatin accessibility in vivo, we performed ATAC-seq analysis for the hippocampus of *Chd8* CKO and control mice after vehicle treatment or of those with a similar seizure stage at 60 min after KA treatment. Most ATAC-seq peaks (51.6 to 74.0% of total peaks) were shared among these four conditions (Fig. 4d). Heat map and density profiles around high-confidence CHD8 ChIP-seq peaks revealed a small decrease in ATAC-seq signal intensity in the hippocampus of *Chd8* CKO mice compared with that of control mice after KA or vehicle treatment (Fig. 4e, Supplementary Fig. 7a). Chromatin accessibility at the gene body of several activity-dependent genes including *Fosb*, *Nr4a1*, and *Egr1* was increased by KA treatment in both the control and *Chd8* CKO hippocampus (Fig. 4f, Supplementary Fig. 7b). Furthermore, ATAC-seq signal intensity at these loci was decreased in the hippocampus of *Chd8* CKO mice compared with that of control mice after KA treatment (Fig. 4f, Supplementary Fig. 7b). These results thus indicated that CHD8 is essential for the transcription of and chromatin accessibility at activity-dependent genes in activated neurons in vivo.

**Behavioral phenotypes of mice with induced CHD8 deficiency in adulthood.** Given that dysregulation of activity-dependent gene expression is associated with memory-formation and psychiatric

disorders[5,6], we performed several behavioral tests with *Chd8* CKO and control male and female mice. In the Morris water-maze test, the visible and hidden platform trials over five consecutive days revealed that escape latency was similar in *Chd8* CKO and control mice (Fig. 5a). After training in the visible and hidden platform trials, we performed a probe trial in which the hidden platform was removed from the water maze. The number of target platform crossings did not differ between *Chd8* CKO and control male or female mice (Fig. 5b). The time spent in the target quadrant was significantly increased relative to that in each of the other quadrants for both *Chd8* CKO and control mice (Fig. 5c), suggestive of normal learning and memory formation in *Chd8* CKO mice.

We next examined anxiety-like behavior and abnormal social behavior typical of ASD model mice, including CHD8-haploinsufficient mice[12–15]. Total distance traveled in the open-field test did not differ between genotypes, suggestive of normal locomotor activity in *Chd8* CKO mice (Fig. 6a). Whereas the time spent in the center of the open field was decreased for male *Chd8* CKO mice compared with control mice, that for female *Chd8* CKO mice was similar to that for control mice (Fig. 6b). We did not detect a difference between genotypes for time spent in the open arms during the elevated plus-maze test or for that spent in the light room during the light-dark transition test (Fig. 6c, d). Male and female *Chd8* CKO mice also showed normal self-grooming behavior (Fig. 6e). Female, but not male, *Chd8* CKO mice showed an increase in total contact time during the reciprocal social-interaction test (Fig. 6f). The number of social contacts during this test did not differ between *Chd8* CKO and control mice (Fig. 6g). In the three-chamber sociability test, both *Chd8* CKO and control animals showed a significant preference for a novel mouse (stranger 1) (Fig. 6h, i). In the social-novelty preference test, both *Chd8* CKO and control male mice also showed a significant preference for a novel mouse (stranger 2) over a familiar mouse (stranger 1), whereas female mice of either genotype did not show such a preference (Fig. 6j, k). These results thus suggested that *Chd8* ablation in the adult brain has selective effects on behavioral characteristics.

**Discussion**

We have here shown that homozygous deletion of *Chd8* in mouse postmitotic neurons alters the expression of activity-dependent genes. We also found that *Chd8* expression in the adult brain is not essential for seizure susceptibility or for memory formation in the Morris water-maze test, but is required for the transcription of activity-dependent genes associated with KA-induced seizures. Our data thus provide insight into the role of CHD8 in activity-dependent transcriptional responses in postmitotic neurons.

CHD8 has previously been shown to regulate the proliferation and differentiation of many types of stem and precursor cells through transcriptional control[18–21,24–26]. Although the expression of *Chd8* persists in the adult brain[27], the functional role of CHD8 in postmitotic neurons has been unclear. Our results now show that

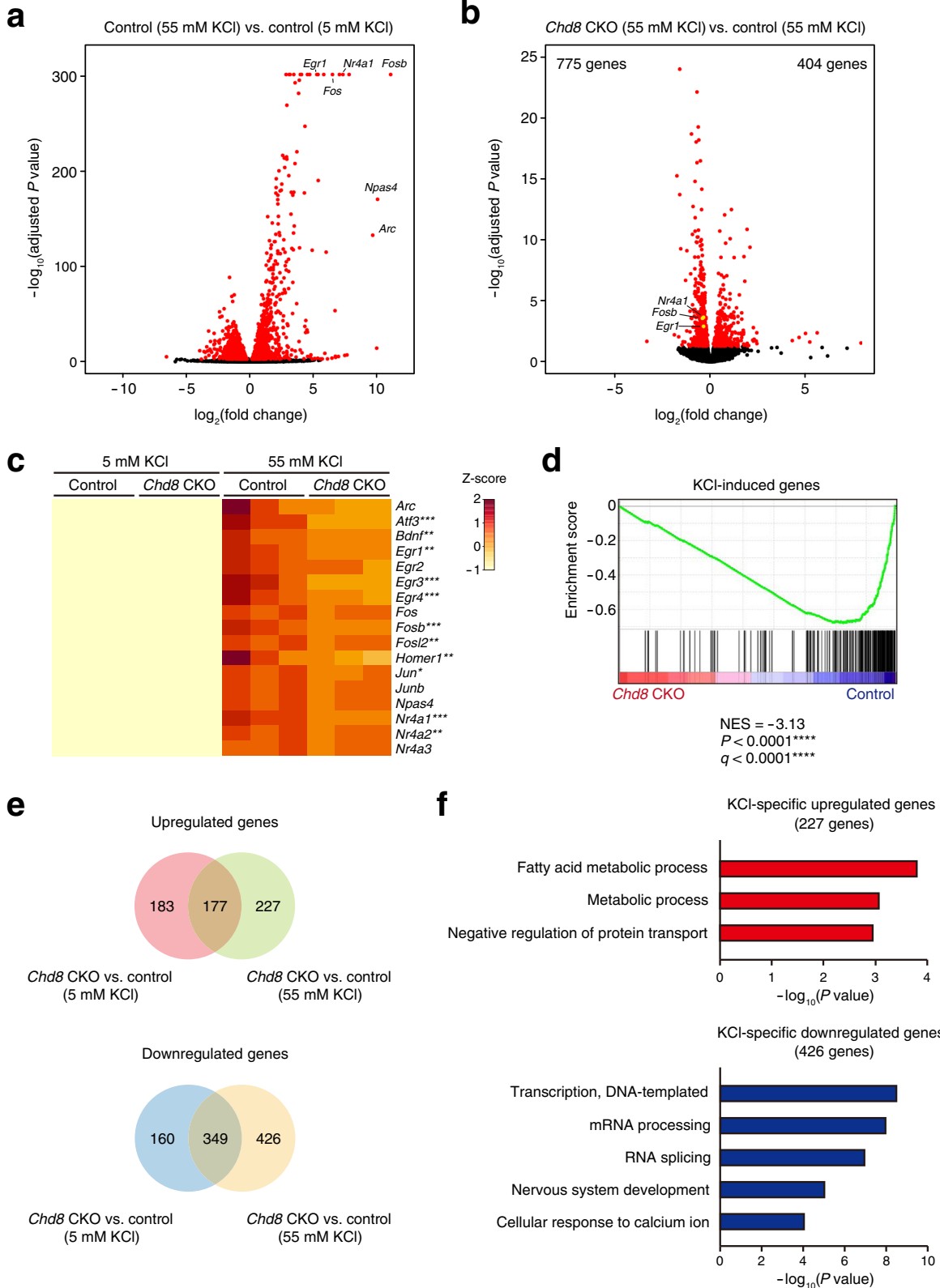

CHD8 contributes to transcriptional regulation in response to neuronal activity. Activity-dependent gene expression mediates synaptic development and plasticity underlying learning and memory[6], and its disruption is related to neuropsychiatric disorders such as ASD[5]. Whereas we detected selective behavioral abnormalities including increased anxiety-like behavior in the open-field test in male *Chd8* CKO mice and abnormal social behavior in female *Chd8* CKO mice in this study, *Chd8* heterozygous knockout mice manifest alterations in synaptic transmission, ASD-like behavioral phenotypes, and learning and memory deficits[12–15,28]. Our observations thus suggest that defective activity-dependent transcriptional regulation by CHD8 may contribute to the neurological phenotypes of individuals with ASD associated with *CHD8* mutations. In addition to the changes in the

**Fig. 2 Activity-dependent gene expression is attenuated by *Chd8* ablation in postmitotic neurons in vitro. a** Volcano plot for differentially expressed genes in neurons isolated from control mice and treated with 55 mM KCl versus 5 mM KCl ($n = 3$ mice per condition). Differentially expressed genes (FDR-adjusted *P* value of <0.05) are highlighted in red. **b** Volcano plot for differentially expressed genes in *Chd8* CKO neurons compared with control neurons under the 55 mM KCl condition ($n = 3$ mice of each genotype, 1 male and 2 females for *Chd8* CKO mice and 2 males and 1 female for control mice). Differentially expressed genes (FDR-adjusted *P* value of <0.05) are highlighted in red. **c** Heat map representing the expression of activity-dependent genes in *Chd8* CKO and control neurons under the 5 and 55 mM KCl conditions ($n = 3$ mice for each condition). Differentially expressed genes in *Chd8* CKO neurons compared with control neurons under the 55 mM KCl condition are indicated by asterisks. **d** GSEA plot of differentially expressed genes among genes induced by KCl (upregulated genes with a $\log_2$(fold change) of >2 associated with an FDR-adjusted *P* value of <0.01 in neurons of control mice treated with 55 mM KCl compared with those treated with 5 mM KCl in **a**) for *Chd8* CKO neurons compared with control neurons under the 55 mM KCl condition. NES, normalized enrichment score. **e** Venn diagrams showing the overlap between genes whose expression was upregulated or downregulated in *Chd8* CKO neurons compared with control neurons under the 5 and 55 mM KCl conditions. **f** GO analysis of genes whose expression was specifically upregulated (227 genes) or downregulated (426 genes) in *Chd8* CKO neurons treated with 55 mM KCl as indicated in **e**. The significance levels for the values of *P* and FDR *q* are indicated by * for <0.05, ** for <0.01, *** for <0.001, and **** for <0.0001.

expression of activity-dependent genes, we found that genes related to translation were highly enriched among downregulated genes in *Chd8* CKO neurons. Local translation in neurons supplies proteins to axons and synapses that are required for synaptic plasticity and neuronal function[29]. Dysregulation of translation has previously been implicated in ASD-like phenotypes[30]. CHD8 thus contributes to transcriptional regulation related to multiple processes including transcription, translation, and nervous system development, and defective regulation of these processes associated with *CHD8* mutation might underlie ASD pathogenesis.

Neuronal activity dynamically modifies the accessible chromatin landscape in association with activity-dependent gene expression in the adult brain[31]. In contrast, our ATAC-seq analysis revealed only small changes in chromatin accessibility in cultured neurons exposed to an elevated extracellular concentration of KCl, similar to previous findings[32]. These differences in the effects of neuronal activity on chromatin accessibility may be due to differences in experimental conditions between neurons in vivo and those in culture. Indeed, our ATAC-seq analysis of the hippocampus of KA-treated mice revealed an increase in chromatin accessibility at the gene body of activity-dependent genes. Consistent with previous findings that CHD8 promotes the adoption of open chromatin structures in several cell types[19,20,33], we found that *Chd8* ablation in the adult brain reduced chromatin accessibility at the loci of activity-dependent genes. Although we did not detect chromatin accessibility changes in response to *Chd8* deletion in vitro, likely as a result of technical limitations, it is possible that CHD8 also affects chromatin accessibility in cultured neurons to some extent, a possibility that warrants further investigation. Given that CHD8 binds to the promoter region of activity-dependent genes, it may promote chromatin accessibility at the regulatory elements of these genes and thereby facilitate their activity-dependent transcription. Further studies will be needed to clarify the detailed mechanisms of transcriptional regulation by CHD8.

CHD8 haploinsufficiency is a highly penetrant risk factor for ASD, and *Chd8* heterozygous mutation confers ASD-like behavioral phenotypes in mice[7,12–17]. Some individuals with ASD associated with *CHD8* mutation experience seizures[7], whereas we did not detect a change in seizure susceptibility in *Chd8* CKO mice. Abnormal social behavior, such as an increased total contact time during the reciprocal social-interaction test, is one of the reproducible behavioral features of *Chd8* mutant mice[12,13,16,17,21]. Female, but not male, *Chd8* CKO mice in the present study showed altered social behavior, suggestive of a sexually dimorphic phenotype[15]. Furthermore, *Chd8* CKO male mice manifested increased anxiety-like behavior in the open-field test, but not in the elevated plus-maze test and the light-dark transition test. Anxiety is one of the symptoms of individuals with ASD[1,7], and *Chd8* heterozygous mutant mice also manifest anxiety-like

behavior[12,13]. Our results suggest that CHD8 in the adult brain may contribute to the control of social behavior and anxiety-like behavior.

Sexually dimorphic behavioral phenotypes have previously been observed in ASD model mice including *Chd8* mutant mice[15]. Given that sex hormones and sex chromosomes are implicated in sex differences in behavior[34], it is possible that *Chd8* deletion alters the levels of sex hormones or the expression of their receptors. Genetic background, age, and mutation differences may also influence sexually dimorphic behavioral outcomes[35,36].

We used the *CAG-CreER* mouse line to achieve tamoxifen-inducible *Chd8* deletion in mouse postmitotic neurons in vitro and in the adult brain in vivo. Given that this line drives recombination in all cell types, rather than showing cell type specificity, *Chd8* deletion not only in postmitotic neurons but in glial cells and other cell populations might influence transcriptomic, epigenetic, and behavioral changes observed in this study. Further studies are warranted to determine whether these changes are attributable to alterations in postmitotic neurons.

CHD8 was previously shown to have dosage-sensitive roles in transcriptional regulation and behavioral phenotypes[33,37]. Given that homozygous deletion of *Chd8* in the adult brain resulted in transcriptional alterations and some behavioral deficits in the present study, it is also possible that *Chd8* heterozygous mutation might confer similar but less pronounced effects. Collectively, the present findings reveal a role for CHD8 in activity-dependent transcriptional regulation in postmitotic neurons and the adult brain, and they therefore provide potentially important insight into the molecular mechanisms underlying the pathogenesis of ASD.

## Methods

**Mice.** Generation of *Chd8*^F/F^ mice was described previously[12]. *Chd8*^F/F^ mice were crossed with *CAG-CreER* heterozygous mice to produce *CAG-CreER/Chd8*^F/F^ mice[24]. For induction of Cre-mediated recombination in vivo, *CAG-CreER/Chd8*^F/F^ mice at 8 to 12 weeks of age were injected intraperitoneally for five consecutive days with tamoxifen (2 mg per mouse) dissolved in corn oil. Multiple treatment with tamoxifen results in more efficient deletion of floxed alleles compared with a single treatment[38]. Mice were genotyped by PCR-based analysis of genomic DNA with primers for *Chd8* (5′-CCCAAAAGACCAAATCAAACAAAC-3′, 5′-CCATA GGCTGAAGAACCGTAATTG-3′, and 5′-AGGCTTAGAAACCCGTCGAG-3′) and *Cre* (5′-AGGTTCGTTCACTCATGGA-3′ and 5′-TCGACCAGTTTAGTTA CCC-3′). All experiments were approved by the Animal Ethics Committee of Kanazawa University.

**Primary neuronal culture.** Primary neurons were isolated from the hippocampus of male and female mice at E18.5 as previously described, but with minor modifications[23,39]. The generation of pyramidal neurons is almost complete at this stage, and isolation of neurons from embryonic tissue has the advantages that the tissue is more readily dissociated and that contamination with glial cells and fibroblasts is minimized. In brief, the tissue was incubated for 20 min at 37 °C with 0.25% trypsin–EDTA and DNase (047-26771, Wako) at 25 μg/ml, and it was subjected to gentle dissociation by repeated passage through a Pasteur pipette after

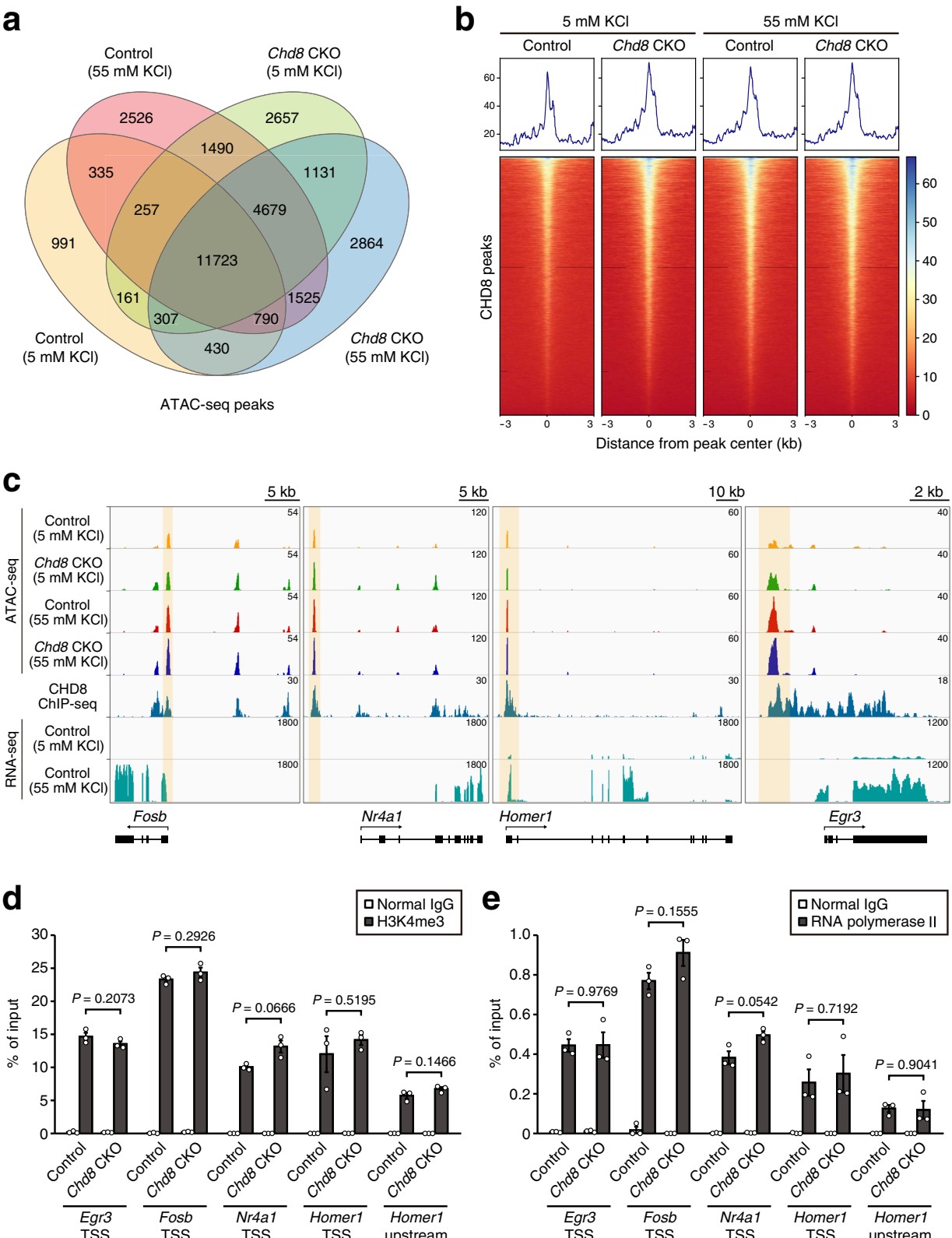

the addition of Dulbecco's modified Eagle's medium (DMEM) supplemented with 10% fetal bovine serum and DNase (10 μg/ml). Dissociated cells were passed through a 40-μm cell strainer, collected by centrifugation, and transferred to a plate coated with poly-D-lysine (P6407, Sigma-Aldrich) for culture at 37 °C in neurobasal medium supplemented with MACS NeuroBrew-21 supplement (130-097-263, Miltenyi Biotec) at 20 ml/l, 2 mM L-glutamine (25030081, Thermo Fisher Scientific), and penicillin-streptomycin (15140122, Thermo Fisher Scientific) at 10 ml/l. After 2 days in vitro (DIV), the cultures were treated with 1 μM Ara-C for 24 h to

eliminate all dividing cells. For the induction of Cre-mediated recombination in vitro, neurons were incubated for 24 h in the presence of 500 μM 4-OHT at 4 DIV. For KCl-induced neuronal depolarization, 1 μM tetrodotoxin (Wako) and 100 μM D-AP5 (Tocris Bioscience) were added to the culture medium at 9 DIV to reduce neuronal activity and the neurons were then treated with 5 or 55 mM KCl for 2 h by adding control buffer (5 mM KCl, 165 mM NaCl, 1.8 mM CaCl$_2$, 0.8 mM MgCl$_2$, 11 mM HEPES) or depolarization buffer (170 mM KCl, 1.8 mM CaCl$_2$, 0.8 mM MgCl$_2$, 11 mM HEPES) to a final dilution of 32.4% at 10 DIV.

**Fig. 3 Chromatin accessibility profile at CHD8 binding sites for neurons activated by KCl-induced depolarization. a** Venn diagram showing the overlap between ATAC-seq peaks detected in *Chd8* CKO or control neurons isolated from male mice and exposed to 5 or 55 mM KCl ($n = 2$ mice per condition). Merged data for two independent ATAC-seq replicates for each condition were used for analysis. The separate results of the two replicates for each condition are shown in Supplementary Fig. 5. **b** Grouping of signal density and heat maps of ATAC-seq peaks in the region spanning 3 kb upstream to 3 kb downstream of the center of CHD8 binding peaks. CHD8 ChIP-seq data are from a previous study[12]. **c** ATAC-seq, CHD8 ChIP-seq, and RNA-seq data for representative activity-dependent genes viewed in the Integrative Genomics Viewer browser. The yellow shaded areas indicate prominent overlapping ATAC-seq and ChIP-seq peaks. **d**, **e** ChIP-qPCR analysis was performed for H3K4me3 deposition (**d**) and RNA polymerase II binding (**e**) at the TSS region of the indicated activity-dependent genes (or at the upstream region of *Homer1* examined as a negative control) in hippocampal neurons isolated from *Chd8* CKO or control mice and treated with 55 mM KCl for 2.0 h. ChIP was performed with normal immunoglobulin G (IgG) as a control for the antibodies to H3K4me3 or to RNA polymerase II. Data are means ± s.e.m. ($n = 3$ independent experiments). The *P* values were determined with the unpaired Student's *t* test.

**Antibodies**. Rabbit polyclonal antibodies to CHD8 were generated in-house and used for immunoblot analysis. Other antibodies included those to TUBB3 (ab78078, Abcam, 1:500), to GFAP (IR524, DAKO, 1:500), and to Olig2 (AB9610, Millipore, 1:500) for immunofluorescence staining, those to FOSB (ab184938, Abcam, 1:2000) and to HSP90 (610419, BD Biosciences, 1:2000) for immunoblot analysis, and those to H3K4me3 (ab8580, Abcam, 2 µg per sample) and to RNA polymerase II (91151, Active Motif, 2 µg per sample) for ChIP.

**Immunofluorescence staining**. Immunofluorescence staining was performed as described previously[21]. The cultured cells were fixed overnight at 4 °C with 4% paraformaldehyde in phosphate-buffered saline (PBS), exposed to 2% bovine serum albumin and 0.3% Triton X-100 in PBS, and then incubated overnight at 4 °C with primary antibodies. Immune complexes were detected with Alexa Fluor 488– or Alexa Fluor 546–conjugated goat secondary antibodies (Thermo Fisher Scientific). The TUNEL (terminal deoxynucleotidyl transferase dUTP nick-end labeling) assay was performed with the use of a MEBSTAIN Apoptosis TUNEL Kit Direct (8445, MBL). All cells were counterstained with 4′,6-diamidino-2-phenylindole (DAPI), and images were acquired with a BZ-X800 microscope (Keyence). ImageJ software was applied to count the numbers of each cell type or apoptotic cells.

**RT and real-time PCR analysis**. Total RNA (500 ng) isolated from cultured neurons or the hippocampus with the use of the TRIzol reagent (Thermo Fisher Scientific) was subjected to RT with ReverTra Ace RT mix with gDNA remover (Toyobo). The resulting cDNA was subjected to real-time PCR analysis with the use of Luna Universal qPCR Master Mix (M3003, New England Biolabs) and specific primers in a Thermal Cycler Dice Real Time System III (Takara Bio). Data were normalized by the abundance of *Rplp0* or *Gapdh* mRNA. The PCR primers (sense and antisense, respectively) were as follows: *Rplp0*, 5′-GGACCCGAGAAGA CCTCCTT-3′ and 5′-GCACATCACTCAGAATTTCAATGG-3′; *Gapdh*, 5′-GCCT GGAGAAACCTGCCAAGTATG-3′ and 5′-GAGTGGGAGTTGCTGTTGAAGTC G-3′; *Chd8$_L$*, 5′-TCCCTTTTTGGTCATTGCTC-3′ and 5′-TTCAGCCTATGGGC TTCATC-3′; *Fosb*, 5′-TTTTCCCGGAGACTACGACTC-3′ and 5′-GTGATTGCG GTGACCGTTG-3′; *Nr4a1*, 5′-TTGAGTTCGGCAAGCCTACC-3′ and 5′-GTGTA CCCGTCCATGAAGGTG-3′; and *Egr1*, 5′-TCGGCTCCTTTCCTCACTCA-3′ and 5′-CTCATAGGGTTGTTCGCTCGG-3′.

**Immunoblot analysis**. Total protein extracts were prepared from cultured neurons or the hippocampus and subjected to immunoblot analysis as previously described[40]. ImageJ software was applied to measure the signal intensity for each protein.

**RNA-seq analysis**. Total RNA was extracted from neurons treated with 5 or 55 mM KCl with the use of the TRIzol reagent. Messenger RNA (1 µg) purified from the total RNA with the use of a NEBNext Poly(A) mRNA Magnetic Isolation Module (New England Biolabs) was used to prepare a cDNA library with the use of a NEBNext Ultra II Directional RNA Library Prep Kit for Illumina (New England Biolabs), and each library was then sequenced with the use of a NovaSeq 6000 system (Illumina). The quality of the raw sequencing data was checked with FastQC (version 0.11.9), and trimming of the adapter sequences was performed with Trimmomatic (version 0.39)[41]. The total amount of each mRNA was calculated with the use of a series of programs including HISAT2 (version 2.1.0)[42], featureCounts (version 2.0.0)[43], and DESeq2 (version 1.26.0)[44]. RNA-seq reads were mapped against the mouse (mm10) genome. GSEA was performed as described previously with the use of GSEA software version 4.2.1[45]. A set of genes whose expression was significantly upregulated (log$_2$(fold change) of >2.0 associated with an FDR-adjusted *P* value of <0.01) in control neurons treated with 55 mM KCl relative to those treated with 5 mM KCl was used for GSEA. GO analysis of differentially expressed genes (FDR *q* < 0.05) was performed with the use of DAVID[46] and SynGO[47].

**ATAC-seq analysis**. ATAC-seq libraries were prepared with the use of an ATAC-Seq Kit (Active Motif). *Chd8* CKO and control neurons isolated from male mice were treated with 5 or 55 mM KCl for 2 h and then incubated for 30 min at 37 °C in culture medium containing DNase (15 µg/ml). The neurons were then dissociated from the culture plate by exposure to 0.25% trypsin–EDTA. The hippocampus was manually dissociated from the brain of mice of each genotype with a similar seizure stage at 60 min after KA or vehicle treatment. Nuclei were extracted from the cultured cells or hippocampus with the use of ATAC Lysis Buffer, and $1 \times 10^5$ nuclei were used to prepare each ATAC-seq library. The libraries were sequenced with the use of a HiSeq 2500 system (Illumina). The reads were uniquely mapped to the mouse (mm10) genome with the use of Bowtie software (version 2.2.3)[48], and duplicated reads were removed with samtools (version 1.9)[49]. BAM files for two replicates for each condition were merged with samtools. Markedly enriched regions of the genome were identified with the use of the MACS peak caller (version 2.1.1, with the option "-p 1e$^{-5}$ --gsize mm --nomodel --extsize 160")[50]. Heat map and density profiles were generated with plotHeatmap in deepTools (3.5.0)[51].

**ChIP-qPCR assay**. ChIP was performed essentially as described previously[21]. Cultured neurons treated with 55 mM KCl for 2 h were fixed by incubation for 10 min on ice with 0.5% paraformaldehyde in ChIP buffer (5 mM HEPES-KOH (pH 8.0), 200 mM KCl, 1 mM CaCl$_2$, 1.5 mM MgCl$_2$, 5% sucrose, 0.5% Nonidet P-40) supplemented with a protease inhibitor cocktail (Wako), subjected to ultrasonic treatment, and digested with micrococcal nuclease for 40 min at 30 °C. After the addition of EDTA to a final concentration of 0.1 mM, each digested sample was centrifuged at $15,000 \times g$ for 10 min at 4 °C, and the resulting supernatant was incubated with rotation for 6 h at 4 °C with antibodies conjugated to magnetic beads. Bound proteins were eluted from the beads, and cross-links were reversed by incubation overnight at 65 °C with 1% SDS in Tris-EDTA buffer. After washing twice both with ChIP buffer and with Tris-EDTA buffer, DNA was purified with the use of Nucleo Spin Gel and PCR Clean-Up (Takara Bio) and subjected to real-time PCR analysis as described above. PCR primers (sense and antisense, respectively) were as follows: *Egr3* TSS, 5′-GGAAGGCTTGGTTGGAGAC-3′ and 5′-GC ACCTACCTCCCTCCAGTC-3′; *Fosb* TSS, 5′-AGCCTGGACTTTCAGGAGGT-3′ and 5′-GCTCGGGGAAGCTTAGTCTC-3′; *Nr4a1* TSS, 5′-AACCTGCACTGG GGTATCAC-3′ and 5′-GACAAAGCTTGGCTTCCTTG-3′; *Homer1* TSS, 5′-GC CTTTAGGAGGGGAGAAAG-3′ and 5′-GGGGAAAACCACCGTTAAT-3′; and *Homer1* upstream, 5′-TCTGCCACCTCATTTCTGCT-3′ and 5′-TAGCACACAC AGGCCATCAT-3′.

**Analysis of published ChIP-seq data**. The ChIP-seq data previously obtained with antibodies to CHD8 (DRA003116) were reanalyzed as described in the original study[12]. In brief, the reads were uniquely mapped to the mouse (mm10) genome with the use of Bowtie software (version 2.2.3)[48], and duplicated reads were removed with samtools (version 1.9)[49]. Markedly enriched regions of the genome were identified with the use of the MACS peak caller (version 2.1.1, with the option "-p 1e$^{-5}$ --gsize mm --nomodel --extsize 160")[50].

**Induction of seizures with KA**. Male mice at 16 to 18 weeks of age and female mice at 13 to 18 weeks of age were injected intraperitoneally with KA (25 mg/kg) dissolved in PBS. Induction of seizures by KA is one of the most commonly studied models of temporal lobe epilepsy[52]. Behavioral seizures were observed for 1 h after the injection and were scored according to previously described criteria[53]: stage 0, normal behavior; stage 1, immobility and rigidity; stage 2, head bobbing; stage 3, forelimb clonus and rearing; stage 4, continuous rearing and falling; stage 5, clonic-tonic seizure; stage 6, death. The hippocampus was then removed, flash frozen in liquid nitrogen, and stored at –80 °C for analysis of gene expression. Mice with a seizure stage of 3 to 5 were used for gene expression analysis and ATAC-seq analysis.

**General protocol for behavioral tests**. *Chd8* CKO or control mice were group-housed in a room with a 12-h-light, 12-h-dark cycle (lights on at 8:00 a.m.) and with access to food and water ad libitum. Behavioral tests were performed with male and female mice at 11 to 18 weeks of age and between 9:00 a.m. and 6:00 p.m. as described previously[19]. Each apparatus was cleaned with a dilute sodium hypochlorite solution before testing of each animal in order to prevent bias due to

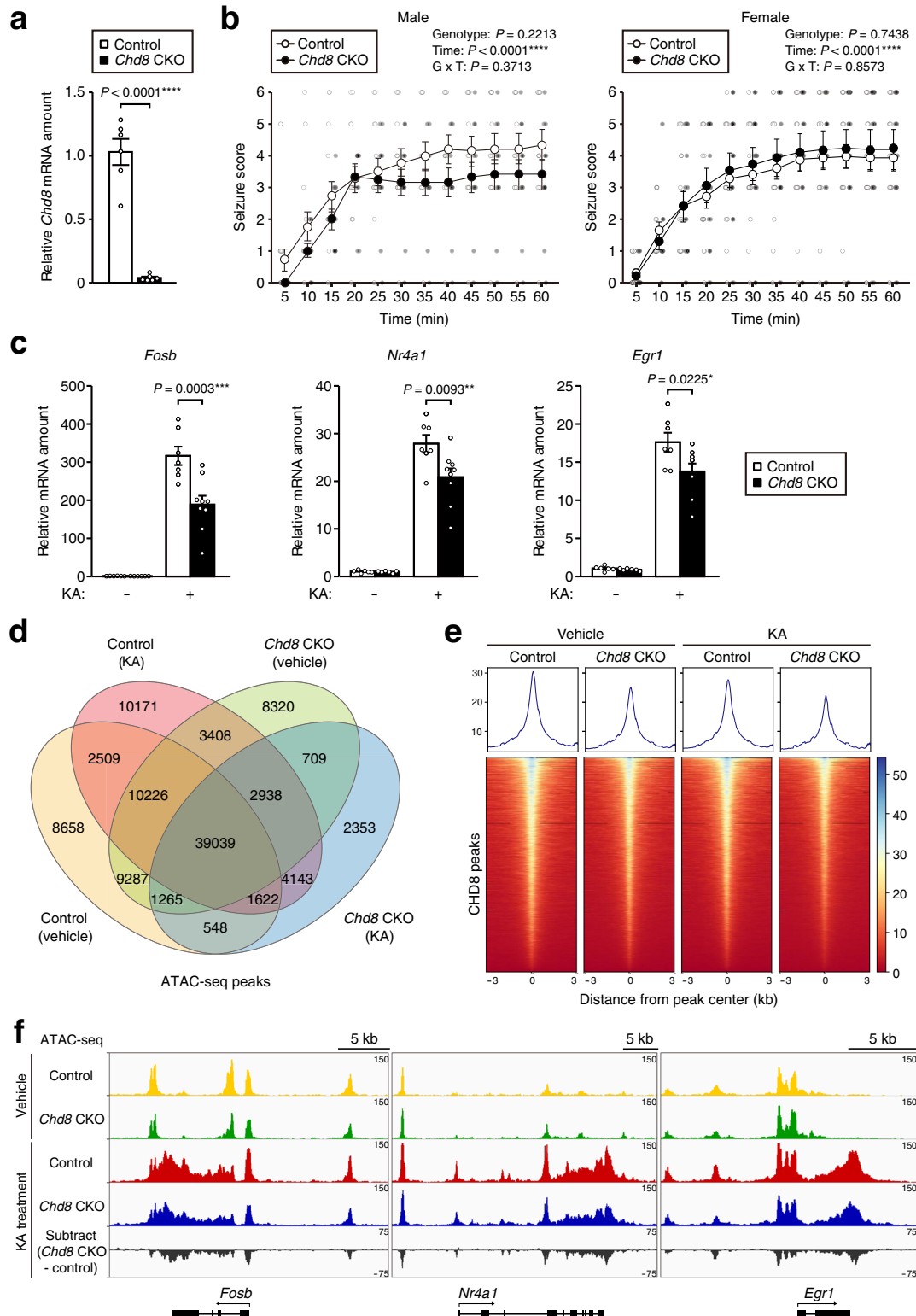

olfactory signals. Behavioral tests included the Morris water-maze test, open-field test, light-dark transition test, elevated plus-maze test, self-grooming test, social-interaction test in a novel environment, and sociability and social-novelty pre-ference tests. All tests were carried out by well-trained experimenters using auto-mated analysis systems as described below. The experimenters were always blinded to mouse genotype in order to exclude bias in behavioral measurements.

**Morris water-maze test**. For visible and hidden platform trials of the Morris water-maze test, a circular plastic pool (120 cm in diameter) was filled to a depth of 30 cm with water (maintained at 21° ± 1.0 °C) that had been colored white by the addition of nontoxic paint[54]. A transparent circular escape platform (9 cm in diameter) was submerged in the center of one of the pool quadrants with its surface located ~1 cm below that of the water. Visual cues that differed in geometric shape and color were distributed around the pool. Male mice between 11 and 15 weeks of age and female mice between 11 and 16 weeks of age were initially subjected to the visible platform task, in which the platform was marked with a flag. The next day, the flag was removed, and the hidden platform tasks were performed on four consecutive days. The mouse was randomly placed in the pool facing the wall in each of the four quadrants. Each task consisted of four trials per day with a cutoff time of 60 s. If the mouse reached the target platform, it was allowed to remain on the platform for >15 s.

**Fig. 4 Activity-dependent gene expression is attenuated by *Chd8* ablation in the adult brain in vivo. a** RT-qPCR analysis of *Chd8* mRNA in the hippocampus of *CAG-CreER/Chd8*^F/F^ (*Chd8* CKO) and *Chd8*^F/F^ (control) male mice at 17 to 18 weeks of age after treatment with tamoxifen for five consecutive days at 8 to 12 weeks of age ($n = 6$ mice of each genotype). **b** Time course of seizure score for 60 min after KA injection in *Chd8* CKO ($n = 12$) and control ($n = 14$) male mice between 16 and 18 weeks of age (left panel) or *Chd8* CKO ($n = 18$) and control ($n = 16$) female mice between 13 and 18 weeks of age (right panel). **c** RT-qPCR analysis of mRNA abundance for the indicated activity-dependent genes in the hippocampus of *Chd8* CKO mice treated with vehicle ($n = 6$), control mice treated with vehicle ($n = 6$), *Chd8* CKO mice treated with KA ($n = 9$), and control mice treated with KA ($n = 7$) at 16 to 18 weeks of age. Male mice with a seizure stage of 3 to 5 were studied for this analysis. Data are means ± s.e.m. **d** Venn diagram showing the overlap between ATAC-seq peaks detected in the hippocampus of *Chd8* CKO and control mice with a similar seizure stage at 60 min after KA or vehicle treatment ($n = 2$ mice per condition: 1 male and 1 female for *Chd8* CKO and control mice treated with KA, 2 males for *Chd8* CKO and control mice treated with vehicle). Merged data for two independent ATAC-seq replicates for each condition were used for analysis. The separate results for the two replicates of each condition are shown in Supplementary Fig. 7. **e** Grouping of signal density and heat maps of ATAC-seq peaks in the region spanning 3 kb upstream to 3 kb downstream of the center of CHD8 binding peaks. **f** ATAC-seq signals of representative activity-dependent genes viewed in the Integrative Genomics Viewer browser. *$P < 0.05$, **$P < 0.01$, ***$P < 0.001$, ****$P < 0.0001$ (unpaired Student's *t* test in (**a**), two-way repeated ANOVA in (**b**), and one-way ANOVA with Tukey's post hoc test in (**c**)).

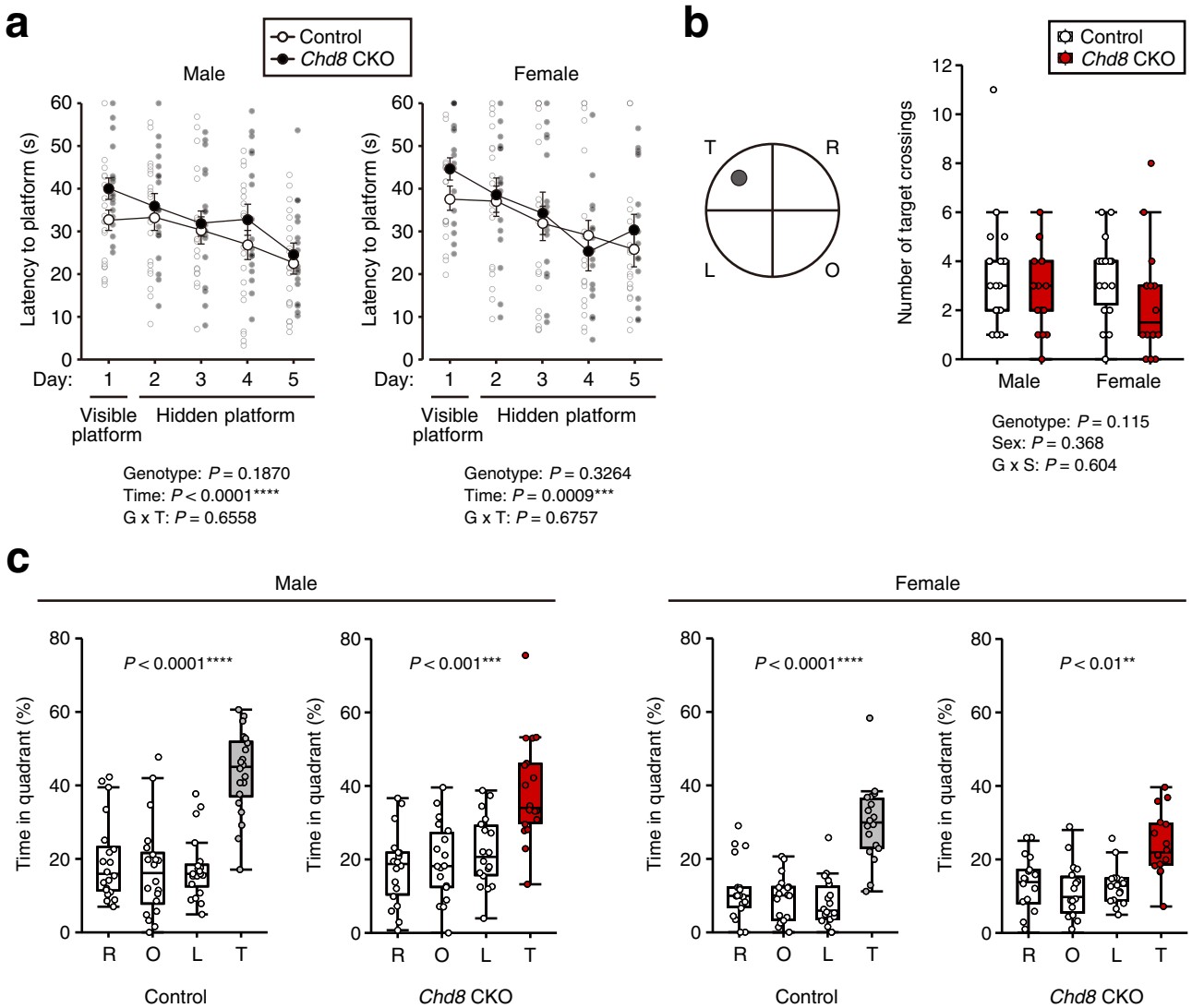

**Fig. 5 Normal learning and memory phenotypes of mice with CHD8 deficiency in adulthood. a** Latency to reach the visible and hidden platforms in the Morris water-maze test for *Chd8* CKO ($n = 18$) and control ($n = 20$) male mice at 11 to 15 weeks of age (left panel) or for *Chd8* CKO ($n = 16$) and control ($n = 18$) female mice between 11 and 16 weeks of age (right panel). Data are means ± s.e.m. **b, c** Number of target crossings (**b**) and time spent in each quadrant (**c**) after platform removal on day 6 for the probe trial of the Morris water-maze test performed with the mice in (**a**). T, O, L, and R represent the target, opposite, left, and right quadrants, respectively. The gray circle represents the target platform. Data are presented as box-and-whisker plots, in which the lower and upper edges of the box indicate the 25th and 75th percentiles, respectively, the central bar indicates the median, and the whiskers indicate nonoutlier extremes. **$P < 0.01$, ***$P < 0.001$, ****$P < 0.0001$ (two-way repeated ANOVA (**a**), two-way factorial ANOVA (**b**), or one-way ANOVA with Tukey's post hoc test (**c**)).

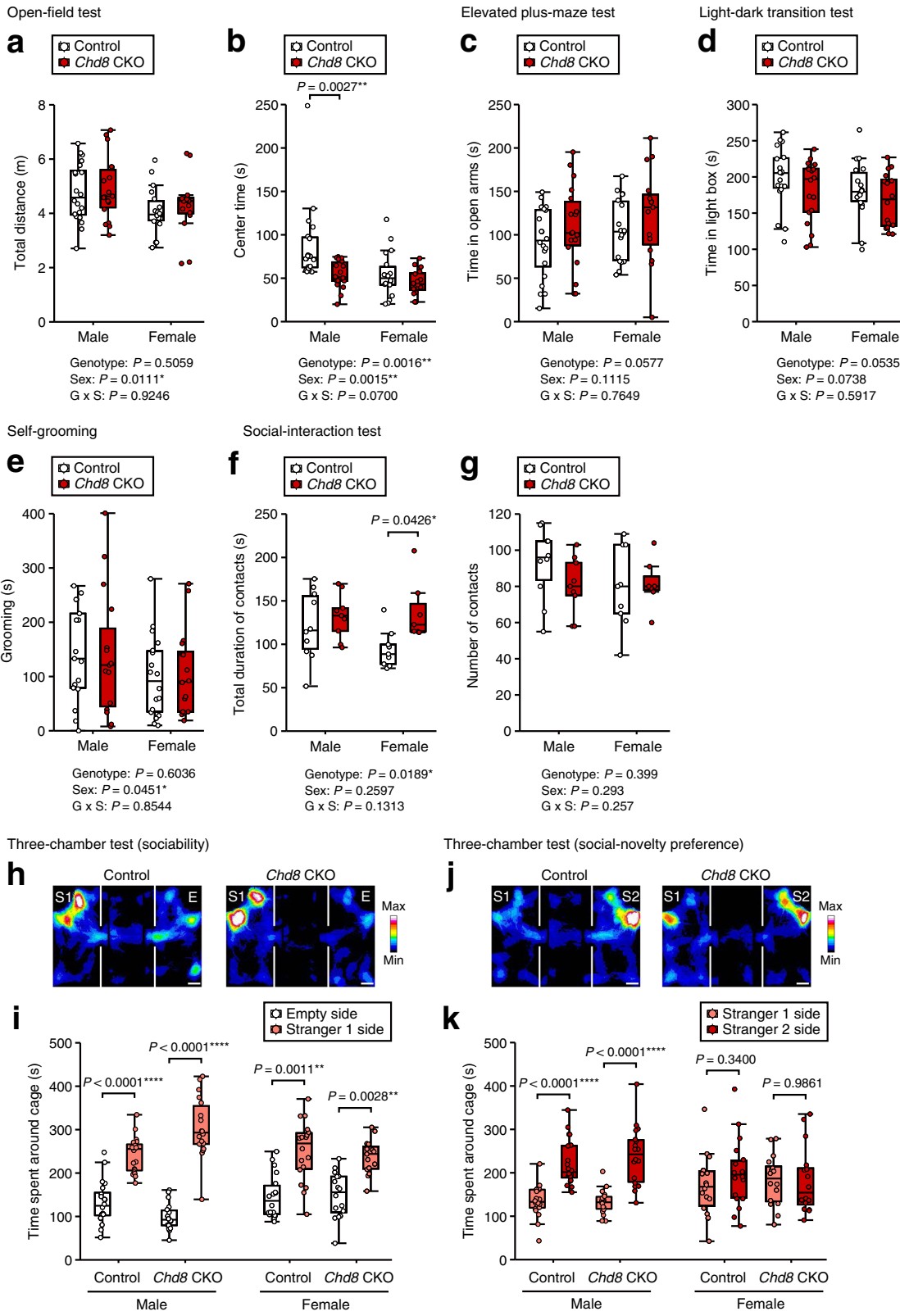

If the mouse did not find the platform within 60 s, it was gently guided to it and then left there for >15 s. On the 6th day, the platform was removed from the pool and a probe trial was performed to assess memory of the previous location of the platform. Mice were allowed to swim in the pool for 60 s, and the time spent in each quadrant was measured. Mouse locomotion was recorded with a video camera and was analyzed automatically with SMART Video Tracking software (Panlab).

**Open-field test**. Each male mouse at 14 to 16 weeks of age or female mouse at 12 to 17 weeks of age was placed in the corner of an open-field apparatus (50 by 50 by 40 cm, O'Hara & Co.), which was illuminated at 100 lux. Total distance traveled and time spent in the central area (25 by 25 cm) were recorded over 10 min. Mouse locomotion was recorded with a video camera controlled with a custom-made program in LabVIEW and was analyzed automatically with in-house software written in Python.

**Fig. 6 Behavioral phenotypes of mice with CHD8 deficiency in adulthood. a, b** Total distance traveled (**a**) and time spent in the central area (**b**) for the open-field test performed with *Chd8* CKO ($n = 18$) and control ($n = 20$) male mice between 14 and 16 weeks of age or with *Chd8* CKO ($n = 16$) and control ($n = 18$) female mice between 12 and 17 weeks of age. **c** Time spent in the open arms for the elevated plus-maze test performed with *Chd8* CKO ($n = 18$) and control ($n = 20$) male mice at 15 to 17 weeks of age or with *Chd8* CKO ($n = 16$) and control ($n = 18$) female mice between 12 and 17 weeks of age. **d** Time spent in the light chamber for the light-dark transition test performed with *Chd8* CKO ($n = 18$) and control ($n = 20$) male mice between 14 and 17 weeks of age or with *Chd8* CKO ($n = 16$) and control ($n = 18$) female mice between 12 and 17 weeks of age. **e** Total duration of self-grooming during a 10-min period for *Chd8* CKO ($n = 16$) and control ($n = 17$) male mice between 11 and 16 weeks of age or for *Chd8* CKO ($n = 16$) and control ($n = 18$) female mice between 13 and 18 weeks of age. **f, g** Total duration of contacts (**f**) and number of contacts (**g**) for the social-interaction test in a novel environment performed with *Chd8* CKO ($n = 9$) and control ($n = 10$) male mice at 15 to 17 weeks of age or with *Chd8* CKO ($n = 7$) and control ($n = 9$) female mice between 12 and 17 weeks of age. **h, i** Representative traces for male mice (**h**) and time spent around each cage (**i**) for the three-chamber sociability test performed with *Chd8* CKO ($n = 18$) and control ($n = 20$) male mice at 16 to 18 weeks of age or with *Chd8* CKO ($n = 16$) and control ($n = 18$) female mice between 13 and 17 weeks of age. Scale bars, 5 cm. **j, k** Representative traces for male mice (**j**) and time spent around each cage (**k**) for the three-chamber social-novelty preference test performed with the mice in **h** and **i**. Scale bars, 5 cm. Data are presented as box-and-whisker plots, in which the lower and upper edges of the box indicate the 25th and 75th percentiles, respectively, the central bar indicates the median, and the whiskers indicate nonoutlier extremes (**a–g**, **i**, **k**). *$P < 0.05$, **$P < 0.01$, ****$P < 0.0001$ (two-way factorial ANOVA with Tukey's post hoc test (**a–g**) or paired Student's *t* test (**i**, **k**)).

**Elevated plus-maze test**. The apparatus consisted of two open arms (25 by 5 cm) and two enclosed arms of the same size with 15-cm-high transparent walls (O'Hara & Co.). The arms and central square were made of white plastic plates and were elevated to a height of 50 cm above the floor. The likelihood of animals falling from the apparatus was minimized by the presence of 3-mm-high plastic ledges on the open arms. Arms of the same type were arranged on opposite sides. Each male mouse at 15 to 17 weeks of age or female mouse at 12 to 17 weeks of age was placed in the central square of the maze (5 by 5 cm) facing one of the closed arms, and its behavior was recorded over 10 min. Mouse locomotion was recorded with a video camera controlled with a custom-made program in LabVIEW, and the time spent in the open arms was measured automatically with in-house software written in Python.

**Light-dark transition test**. The apparatus consisted of a cage (21 by 42 by 25 cm) that was divided into two sections of equal size by a partition with a door (O'Hara & Co.). One chamber was made of white plastic and brightly illuminated (390 lux), whereas the other was black and dark (2 lux). Male mice at 14 to 17 weeks of age or female mice at 12 to 17 weeks of age were placed in the dark side and allowed to move freely between the two chambers with the door open for 10 min. Mouse locomotion was recorded with a video camera controlled with a custom-made program in LabVIEW, and the time spent in each chamber was measured automatically with in-house software written in Python.

**Self-grooming test**. The grooming test was performed as previously described[12]. Each male mouse at 11 to 16 weeks of age or female mouse at 13 to 18 weeks of age was placed individually into a new standard cage. After habituation for 10 min, the animal was videotaped for a 10-min test period and the time spent in grooming behavior was determined.

**Social-interaction test in a novel environment**. Two mice at 15 to 17 weeks of age for males and 12 to 17 weeks of age for females and of the same genotype that had previously been housed in groups of mixed genotypes (three or four animals per cage) and in different cages were placed together in a box (50 by 50 by 40 cm, O'Hara & Co.) and allowed to explore freely for 10 min. Images were captured at a rate of three frames per second, and analysis was performed automatically with in-house software written in Python. The total number of contacts and the total duration of contacts were measured.

**Sociability and social-novelty preference tests**. The testing apparatus consisted of a rectangular three-chambered box (O'Hara & Co.). Each chamber was 20 by 40 by 30 cm, and the dividing walls were made of clear Plexiglas, with small openings (6 cm) that allowed access into each chamber. A sex-matched unfamiliar mouse (stranger 1) that had not had prior contact with the subject mouse was placed in one of the side chambers. The location of stranger 1 in the left versus right chamber was systematically alternated between trials. The stranger mouse was enclosed in a small round wire cage that allowed nose contact between the bars but prevented fighting. The cage was 10 cm in height, with a bottom diameter of 10 cm and vertical bars 0.5 cm apart. An identical empty cage was placed in the other side chamber. The subject mouse was first placed in the middle chamber and was allowed to explore the entire social test box for 10 min. The amount of time spent around each cage and in each chamber was measured with the aid of a camera to quantify the social preference for stranger 1. A second sex-matched unfamiliar mouse (stranger 2) was then placed in the empty cage. The test mouse thus had a choice between the first, already-investigated unfamiliar mouse (stranger 1) and the novel unfamiliar mouse (stranger 2). The amount of time spent around each cage and in each chamber during a second 10-min session was measured as before. C57BL/6J mice were used as stranger mice, and male and female subject mice used in these tests were 16 to 18 weeks or 13 to 17 weeks of age, respectively. Mouse locomotion was recorded with a video camera controlled with a custom-made program in LabVIEW and was analyzed automatically with in-house software written in Python.

**Statistics and reproducibility**. Quantitative data are presented as means ± s.e.m. or as indicated, with the number of mice subjected to each experiment also being stated. Statistical analysis by the unpaired Student's *t* test, the paired Student's *t* test, one-way analysis of variance (ANOVA) with Tukey's post hoc test, two-way factorial ANOVA, or two-way repeated ANOVA was performed with the use of R language. Significance levels for *P* and FDR *q* values are indicated by *, **, ***, and **** for <0.05, <0.01, <0.001, and <0.0001, respectively.

## Data availability

Source data underlying all the graphs is available in the Supplementary Data 2. Data are available by contacting the corresponding author. Uncropped images of the immunoblots are shown in Supplementary Fig. 8. RNA-seq and ATAC-seq data have been deposited in the DDBJ sequence read archive (DRA) under accession numbers DRA014131, DRA016165, and DRA016198.

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

## Acknowledgements

The authors thank M. Asamura, H. Kobayashi, and C. Tambo for overall technical assistance; A. Toyoda (National Institute of Genetics) for technical assistance with RNA-seq and ATAC-seq analysis; K. Ono, T. Hamaguchi, and D. Muramatsu for use of equipment for the Morris water-maze test; T. Miyamoto for general assistance and discussion; and laboratory members for discussion. A.K. was supported by a fellowship from the Japan Society for the Promotion of Science (JSPS). This work was supported in part by KAKENHI grants from JSPS and the Ministry of Education, Culture, Sports, Science, and Technology of Japan to M.N. (JP21H02847 and 16H06279 (PAGS)) and to A.K. (JP21K15726, JP21H05619, and 22H05493) as well as by a PRIME grant from the Japan Agency for Medical Research and Development (AMED) to M.N. (JP22gm6310008).

## Author contributions

A.K. designed and performed experiments, analyzed data, and prepared the manuscript. M.N. contributed to the supervision of the study and writing the manuscript.

## Competing interests

The authors declare no competing interests.
