## [Peer Review File · Communications Biology]

Reviewers' comments:

Reviewer #1 (Remarks to the Author):

This manuscript reports the role of CHD8 in the regulation of the transcription of activity-dependent genes in the brain. The authors use both cultured hippocampal neurons and the adult mouse hippocampus to determine whether an acute/adult deletion of CHD8 leads to changes in the expression of activity-dependent genes. The authors use both RNA-Seq and ATAC-Seq analyses to determine the transcript and chromatin accessibility profiles. The authors also use CHD8-mutant mice to determine the impacts of CHD8 on seizure propensity and various ASD-related behaviors.

Given the emerging importance of the interplay between the loss of ASD-risk genes and various external stimulations, this manuscript is novel in that it explores how CHD8-mutant neurons respond to activity-inducing external stimuli. In addition, the authors find that CHD8 deletion in adulthood can lead to various behavioral deficits, pointing to the important roles of CHD8 expressed during adult stages.

1. Venn Diagram is a useful way to compare transcriptomic changes, but the authors may want to use two-dimensional plotting of two transcript groups so that they do not miss the genes that are significant in one group but only slightly below the significance in another group.
2. For the functional analysis of the transcripts, the authors mainly use GO analysis. It is recommended to use additional analysis such as SynGO, String, and GSEA using all known gene sets (additional to just one gene [KCl-induced genes]).
3. The behavioral assays could be improved in several ways. Open-field locomotor activity results are missing. The exact number of platform crossings was not determined. Light-dark test results could be added to further assess anxiety-like behaviors. Repetitive behavioral data, which is a core ASD-related behavior, are missing.
4. Fig. 5c does not seem to be a representative image.

Reviewer #2 (Remarks to the Author):

Kawamura et al. demonstrates for the first time that CHD8 regulates activity-dependent gene expression in post-mitotic neurons from the hippocampus of male mice, either in vitro in response to KCl or in vivo in response to kainic acid. The mechanism responsible for these deficits remain unknown as no differences in chromatin accessibility was detected in Chd8cKO vs. control neurons.

Although an interesting finding, the study has significant limitations.

- 1) Chd8 deletion has been shown by several groups, including this group, to affect the viability of many cells, including early embryonic cells (Nishiyama, 2009), hematopoietic stem cells (Nota, 2021) and neural progenitors (Hurley et al, 2021). It appears that p53-regulated genes are not induced by CHD8 deletion in post-mitotic neurons, an interesting finding. This should be acknowledged. However, it still remains possible that Chd8 deletion affects the health and viability of cKO neurons. It is important that some data is provided to show that these neurons remain healthy and viable compared to control neurons.
- 2) The authors comment that the lack of inducible chromatin accessibility in response to KCl is seen in vivo, perhaps due to differences between in vitro and in vivo settings. Do the authors see changes in ATACseq in vivo in response kainate or another stimulus as may be expected? If no differences in accessibility is seen in cKO vs. control neurons, what is the mechanism whereby CHD8 deficiency

causes reduced induction of immediate early genes? The authors propose differential recruitment of KMTs and RNAPoIII – for publication in this journal at the very least H3K4me3 and PoII ChIP-qRT-PCR at activity-induced gene promoters would be necessary to confirm or exclude a significant effect on H3K4me3.

3) A significant limitation of this study is that all experiments were performed with male mice - no rationale is provided for this choice and unless a very strong rationale is provided, some data on females should be included.

Minor comments

Line 15-17: CHD8 mutations are highly penetrant but extremely rare, so this statement is inaccurate

Line 42: This statement is a simplification and mis-representation. Not all reported Chd8+/- mice “manifest ASD-like behavioral characteristics” . Robust behavioral studies, some by leading laboratories in the field have been published that did not find any evidence for ASD-like behaviors. Pertinent examples include: Gompers et al. Nat Neurosci 2017, Suetterlin et al, Cereb Cortex 2018, Jimenez et al. Mol Autism 2021. These studies should be cited and the inconsistencies in behavioral findings clearly acknowledged.

Line 155: need to qualify that memory formation was only assessed in the Morris water maze test.

Line 164: should mention the specific behavioral abnormalities found.

Line 190: The studies reporting these findings should be properly cited. Two important studies from leading laboratories in the field that clearly showed these effects are not cited: Suetterlin et al., Cereb Cortex 2018 and Jimenez et al. Mol Autism, 2021.

Reviewer #3 (Remarks to the Author):

Kawamura and Nishiyama report on conditional Chd8 deletion in postmitotic neurons and adult mice in an effort to distinguish postnatal from prenatal consequences of Chd8 mutation. Their focus are activity-dependent processes for which they expose cells to elevated KCl concentrations to induce depolarization and investigate gene transcription and chromatin accessibility. In adult cKO mice, they investigate expression of activity-dependent genes and behavior. While the study is very well conducted and presented the findings remain moderate and mostly confirmatory of previous studies. Importantly, no overt deleterious effects of conditional Chd8 mutation in adult mice can be observed, possibly suggesting that disease relevant abnormalities maybe laid down during prenatal or early postnatal development.

1. I would recommend slightly modifying the diagram in Fig.1a, by making it clear that this is not the process used in all experiments. Maybe TTX and KCl can be added in different color or font, or the diagram can be duplicated with these additions for Fig. 2.

2. I would recommend in experiments using adult mice a breakdown according to sex and also in the cell culture experiments ideally including information on the sex of the embryos used.

3. Interestingly, Chd8 mutant mice show decreased susceptibility to lethal seizures. I would appreciate if the authors could elaborate more on this point. Is this purely the consequence of downregulated expression of activity-dependent genes?

4. I would also appreciate a little more discussion of the identified dysregulated pathways. How well do the pathways in Fig. 1 and 2 align with theories of Chd8-mediated transcriptional control or ASD dysregulation.

Reviewer #4 (Remarks to the Author):

In this manuscript A. Kawamura & M. Nishiyama examine function of the chromatin modifier ASD-related gene Chd8 in postmitotic neurons using both in vitro and in vivo approaches that utilize the well-known Cre recombination system induced by tamoxifen. These authors have published relevant work in the Chd8 field, and this manuscript would nicely add to their Chd8-biology sequelae when appropriately revised. Although the downregulation of expression of neuronal genes as a consequence of Chd8 deletion is expected, and a finding that has been previously described in constitutive Chd8 ablation models; this study is novel in the discovery that links Chd8 deletion specifically in postmitotic neurons with the alteration of the expression of activity-dependent genes induced by KCl-mediated neuronal depolarization. The study is also novel in linking ablation of CHD8 in adult mice with attenuation of activity-dependent transcriptional responses in the hippocampus to kainic acid-induced seizures. Experiments are overall well-controlled and conducted, with some addressable exceptions mentioned below, and findings are mostly presented in a manner that is clear and organized. Overall, the study is somewhat strong but well designed and it should advance to the field of filling gaps in the understanding Chd8 biology. The results are potentially of broad interest to the fields of ASD preclinical models and more generally neurodevelopmental and psychiatric disorders. This reviewer would be happy to review a revised version of this manuscript.

Specific comments/issues, all addressable through adding minimal experimental work and/or acknowledging limitations/deficiencies of the study:

- Please add rationale behind hippocampal in vitro cultures being made at embryonic day E18.5.
- For their in vitro studies, authors consistently refer to primary cultures as neurons culture. Please add any missing methodology or acknowledge through out the manuscript these cultures would also include other cell types (i.e., glia, astrocytes, oligodendrocytes, etc). This is important for assessing Chd8 function in a specific cell type, in this case, and as claimed, in Neurons. Imaging of this neuronal cultures with specific cellular markers would help visualizing the cultures they are working with.
- Please add to the rationale of using KA treatment. Is there a clinical based motivation? Do Chd8 mutation carries present seizures?
- Confirmation of changes at protein levels for the activity dependent genes RNA-Seq findings would validate such changes in expression after the KCl treatment (Arc, Egr1, Fos, Fosb, Npas4, and Nr4a1). Do these changes persist? A time course would be needed to answer this question.
- The claim in line 97: "These results thus suggested that CHD8 is required for the induction of activity-dependent gene transcription in vitro", seems premature at the stage of the finding presented so far; it could be moved to discussion section and expand on the reasoning to conclude this.
- Please expand in the discussion section of the atac-Seq negative findings – what are the limitations of this design?
- Please add the rationale of administering tamoxifen for 5 days to 8–11-week-old mice. Explain why this age and why the 5 days treatment.
- Figure 4a – please add the rationale behind assessing tamoxifen induced Chd8 deletion at the mRNA level only – what is the half-life of the protein? If unknown, then this experiment should include Chd8 protein expression assessed via immunocytochemistry or Western Blot.
- From the disease model perspective, do the heterozygous conditional tamoxifen treated animals exhibit any phenotypes. Were these assessed/tested? Please clarify and explain if heterozygous mice were not analyzed.
- "Although we did not detect a significant difference in seizure severity between Chd8 CKO and control mice after KA treatment, the proportion of animals who achieved seizure stage 6 (death) was reduced for Chd8 CKO mice (1 of 12 mice) compared with control mice (6 of 14 mice) (Fig.4b)." this should be deeply discussed in the discussion section.
- For drawing functional conclusions, in vivo mRNA expression changes (Fosb, Nr4a1, and Egr1) after tamoxifen and KCL/KA treatment should also be confirmed at protein levels. Otherwise acknowledge and explain the reasoning fundamentally.
- Importantly, this manuscript does not consider sex as a biological variable in their in vivo analyses. All the presented data from males. What is the reasoning for this? This should be stated by the

authors in the manuscript acknowledging current field-reporting guidelines, where sex should be considered as a biological variable regardless ASD being more prevalent in males (something that is also changing).

- Last paragraph of the discussion (and overall sense throughout the manuscript): "Collectively, the present findings reveal a role for CHD8 in activity-dependent transcriptional regulation in postmitotic neurons, and they therefore provide potentially important insight into the molecular mechanisms underlying the pathogenesis of ASD". Please discuss on how the behavioral changes observed in a tamoxifen induced Chd8 deletion adult mouse would be relevant to ASD modeling from the point of view of ASD being a developmental disorder. If this aspect of their research is something the authors do not consider relevant to the disease modeling, then the overall angle of the manuscript should change and discussed accordingly in the discussion section.

Response to Reviewer #1

This manuscript reports the role of CHD8 in the regulation of the transcription of activity-dependent genes in the brain. The authors use both cultured hippocampal neurons and the adult mouse hippocampus to determine whether an acute/adult deletion of CHD8 leads to changes in the expression of activity-dependent genes. The authors use both RNA-Seq and ATAC-Seq analyses to determine the transcript and chromatin accessibility profiles. The authors also use CHD8-mutant mice to determine the impacts of CHD8 on seizure propensity and various ASD-related behaviors.

Given the emerging importance of the interplay between the loss of ASD-risk genes and various external stimulations, this manuscript is novel in that it explores how CHD8-mutant neurons respond to activity-inducing external stimuli. In addition, the authors find that CHD8 deletion in adulthood can lead to various behavioral deficits, pointing to the important roles of CHD8 expressed during adult stages.

[Response] We thank the reviewer for the careful review of our manuscript and for the statements that “this manuscript is novel in that it explores how CHD8-mutant neurons respond to activity-inducing external stimuli,” and that “the authors find that CHD8 deletion in adulthood can lead to various behavioral deficits, pointing to the important roles of CHD8 expressed during adult stages.” We also thank the reviewer for suggestions that we feel have helped us to greatly improve our manuscript.

1. Venn Diagram is a useful way to compare transcriptomic changes, but the authors may want to use two-dimensional plotting of two transcript groups so that they do not miss the genes that are significant in one group but only slightly below the significance in another group.

[Response] In response to this comment, we have produced two-dimensional plots (**new Supplementary Fig. 4d–f**) and a table (**new Supplementary Table 4**) for comparisons of $\log_2(\text{fold change})$ and $-\log_{10}(\text{FDR-adjusted } P \text{ value})$ between *Chd8* CKO and control neurons under the 5 and 55 mM KCl conditions. We have now addressed this point in the revised manuscript (page 5, line 105–page 6, line 109).

Kawamura et al. Supplementary Figure 4

2. For the functional analysis of the transcripts, the authors mainly use GO analysis. It is recommended to use additional analysis such as SynGO, String, and GSEA using all known gene sets (additional to just one gene [KCl-induced genes]).

[Response] As suggested by the reviewer, we performed additional functional analysis for our RNA-seq data using SynGO and GSEA. SynGO analysis revealed that downregulated genes in *Chd8* CKO neurons under the 5 and 55 mM KCl conditions were enriched in genes related to pre- and postsynaptic function categories (**new Fig. 1f and Supplementary Fig. 4b**), whereas the upregulated genes showed no significant enrichment (**new Supplementary Figs. 2 and 4a**). Furthermore, GSEA for Kyoto Encyclopedia of Genes and Genomes (KEGG) pathways showed that the expression of ribosomal genes was significantly downregulated in *Chd8* CKO neurons under the 5 and 55 mM KCl conditions (**new Fig. 1g and Supplementary Fig. 4c**), consistent with the results of GO analysis (Fig. 1e). We have now addressed these points in the revised manuscript (page 4, line 79-page 5, line 83; page 5, lines 102-105).

Kawamura *et al.* Supplementary Figure 2

Kawamura *et al.* Figure 1

Kawamura *et al.* Supplementary Figure 4

Kawamura *et al.* Supplementary Figure 4

Kawamura *et al.* Figure 1

Kawamura *et al.* Supplementary Figure 4

3. The behavioral assays could be improved in several ways. Open-field locomotor activity results are missing. The exact number of platform crossings was not determined. Light-dark test results could be added to further assess anxiety-like behaviors. Repetitive behavioral data, which is a core ASD-related behavior, are missing.

[Response] As suggested, we have improved the behavioral data by performing additional behavioral tests. In addition, we have now performed the various behavioral tests with *Chd8* CKO and control female mice in addition to males. Total distance traveled in the open-field test did not differ between genotypes, suggestive of normal locomotor activity in *Chd8* CKO mice (**new Fig. 6a**). We also added the data for the number of target platform crossings in the probe trial of the Morris water-maze test, which did not differ between genotypes (**new Fig. 5b**). Furthermore, we performed the light-dark transition test and self-grooming test to assess anxiety-like and repetitive behaviors, respectively. We did not detect significant differences in the time spent in the light room during the light-dark transition test or in grooming behavior between *Chd8* CKO and control mice (**new Fig. 6d, e**). We have now addressed these issues in the revised manuscript (page 8, lines 163-164; page 8, lines 169-176).

Kawamura *et al.* Figure 5

Kawamura *et al.* Figure 6

4. *Fig. 5c does not seem to be a representative image.*

[Response] To avoid any misunderstanding, we have omitted this image in the revised manuscript.

Response to Reviewer #2

Kawamura et al. demonstrates for the first time that CHD8 regulates activity-dependent gene expression in post-mitotic neurons from the hippocampus of male mice, either in vitro in response to KCl or in vivo in response to kainic acid. The mechanism responsible for these deficits remain unknown as no differences in chromatin accessibility was detected in Chd8cKO vs. control neurons.

Although an interesting finding, the study has significant limitations.

[Response] We thank the reviewer for the careful review of our manuscript and for the statement that “Kawamura et al. demonstrates for the first time that CHD8 regulates activity-dependent gene expression in post-mitotic neurons from the hippocampus of male mice.” We also thank the reviewer for suggestions that we feel have helped us to greatly improve our manuscript.

1) Chd8 deletion has been shown by several groups, including this group, to affect the viability of many cells, including early embryonic cells (Nishiyama, 2009), hematopoietic stem cells (Nita, 2021) and neural progenitors (Hurley et al, 2021). It appears that p53-regulated genes are not induced by CHD8 deletion in post-mitotic neurons, an interesting finding. This should be acknowledged. However, it still remains possible that Chd8 deletion affects the health and viability of cKO neurons. It is important that some data is provided to show that these neurons remain healthy and viable compared to control neurons.

[Response] Phase-contrast and immunofluorescence images showed similar gross morphology and numbers of cells for cultures prepared from *Chd8* CKO and control mice (**new Fig. 1c and Supplementary Fig. 1a, b**). Furthermore, we performed the TUNEL assay to detect apoptotic cells, and found that their frequency did not differ significantly between *Chd8* CKO and control neurons (**new Supplementary Fig. 1c, d**). These results suggest that *Chd8* CKO neurons are healthy and viable, similar to control neurons. We have now addressed this point in the revised manuscript (page 4, lines 68-71).

Kawamura et al. Figure 1

Kawamura *et al.* Supplementary Figure 1

2-1) The authors comment that the lack of inducible chromatin accessibility in response to KCl is seen *in vivo*, perhaps due to differences between *in vitro* and *in vivo* settings. Do the authors see changes in ATACseq *in vivo* in response kainate or another stimulus as may be expected?

[Response] In response to this comment, we performed ATAC-seq analysis for the hippocampus of *Chd8* CKO and control mice with a similar seizure stage at 60 min after KA treatment. Chromatin accessibility at the gene body of several activity-dependent genes including *Fosb*, *Nr4a1*, and *Egr1* was increased by KA treatment, consistent with previous observations (Fernandez-Albert *et al.*, *Nat. Neurosci.* 22: 1718–1730, 2019). We also found that ATAC-seq signal intensity at these loci was decreased in the hippocampus of *Chd8* CKO mice compared with that of control mice (**new Fig. 4d**). These results suggest that changes in chromatin accessibility due to *Chd8* ablation might affect the induction of activity-dependent genes *in vivo*. We have now addressed this issue in the Results section of the revised manuscript (page 7, lines 147-155).

d

Kawamura et al. Figure 4

2-2) If no differences in accessibility is seen in cKO vs. control neurons, what is the mechanism whereby CHD8 deficiency causes reduced induction of immediate early genes? The authors propose differential recruitment of KMTs and RNAPolIII – for publication in this journal at the very least H3K4me3 and PolII ChIP-qRT-PCR at activity-induced gene promoters would be necessary to confirm or exclude a significant effect on H3K4me3.

[Response] As suggested, we examined H3K4me3 deposition and RNA polymerase II recruitment in *Chd8* CKO neurons. ChIP-qPCR analysis revealed that the extent of H3K4me3 and RNA polymerase II enrichment around the transcription start site (TSS) of several activity-dependent genes did not differ between genotypes (**new Supplementary Fig. 5a, b**), suggesting that another mechanism might underlie the transcriptional regulation of these genes by CHD8 *in vitro*. As mentioned above, we found that chromatin accessibility at loci of several activity-dependent genes was decreased by *Chd8* ablation *in vivo* (**new Fig. 4d**). Although we did not detect chromatin accessibility changes in response to *Chd8* deletion *in vitro*, likely due to technical limitations, it is possible that CHD8 might also affect chromatin accessibility *in vitro* to some extent, a possibility that warrants further investigation. We have now addressed these points in the revised manuscript (page 6, lines 127-131; page 7, lines 147-155; page 10, lines 218-226).

Kawamura et al. Supplementary Figure 5

3) A significant limitation of this study is that all experiments were performed with male mice - no rationale is provided for this choice and unless a very strong rationale is provided, some data on females should be included.

[Response] We have now performed behavioral tests with *Chd8* CKO and control female mice as well as males. We did not detect a significant difference in seizure severity between *Chd8* CKO and control female mice after KA treatment (**new Fig. 4b**). In the Morris water-maze test, the visible and hidden platform trials over five consecutive days revealed that escape latency was similar for *Chd8* CKO and control female mice (**new Fig. 5a**). The number of target platform crossings during a probe trial also did not differ between *Chd8* CKO and control female mice (**new Fig. 5b**). Moreover, the time spent in the target quadrant during the probe trial was significantly increased relative to that in each of the other quadrants for both *Chd8* CKO and control female mice (**new Fig. 5c**), suggestive of normal learning and memory formation in *Chd8* CKO mice. Total distance traveled in the open-field test did not differ between genotypes, suggestive of normal locomotor activity in *Chd8* CKO male and female mice (**new Fig. 6a**). Whereas male *Chd8* CKO mice showed a decrease in the time spent in the center of the open field compared with control mice, female *Chd8* CKO mice showed no such difference relative to control mice (**new Fig. 6b**). We did not detect a difference between genotypes of either sex for time spent in the open arms during the elevated plus-maze test or for that spent in the light room during the light-dark transition test (**new Fig. 6c, d**). Male and female *Chd8* CKO mice showed normal self-grooming behavior (**new Fig. 6e**). Female, but not male, *Chd8* CKO mice showed an increase in the total contact time during the reciprocal social-interaction test (**new Fig. 6f**), whereas the number of social contacts did not differ between *Chd8* CKO and control mice of either sex (**new Fig. 6g**). In the three-chamber sociability test, both *Chd8* CKO and control animals of each sex showed a significant preference for a novel mouse (stranger 1) (**new Fig. 6h, i**). In the social-novelty preference test, both *Chd8* CKO and control male mice also showed a significant preference for a novel mouse (stranger 2) over a familiar mouse (stranger 1), whereas female mice of both genotypes did not show a significant preference for a novel mouse (stranger 2) (**new Fig. 6j, k**). These results thus suggested that *Chd8* ablation in the adult brain has selective effects on behavioral characteristics. We have now addressed these points in the revised manuscript (page 7, lines 141-142; page 7, line 157-page 9, line 184).

Kawamura et al. Figure 4

Kawamura *et al.* Figure 5

Open-field test

Elevated plus-maze test

Light-dark transition test

Self-grooming

Social-interaction test

g

Three-chamber test (sociability)

Three-chamber test (social-novelty preference)

Kawamura et al. Figure 6

Minor comments:

-Line 15-17: CHD8 mutations are highly penetrant but extremely rare, so this statement is inaccurate

[Response] We apologize for the inadequate description and have now modified the text in the revised manuscript (page 2, lines 15-16).

-Line 42: This statement is a simplification and mis-representation. Not all reported Chd8+/- mice “manifest ASD-like behavioral characteristics”. Robust behavioral studies, some by leading laboratories in the field have been published that did not find any evidence for ASD-like behaviors. Pertinent examples include: Gompers et al. Nat Neurosci 2017, Suetterlin et al, Cereb Cortex 2018, Jimenez et al. Mol Autism 2021. These studies should be cited and the inconsistencies in behavioral findings clearly acknowledged.

[Response] We apologize for the insufficient description regarding the behavioral characteristics of *Chd8* mutant mice. As pointed out by the reviewer, we and other groups have shown that *Chd8* heterozygous mutant mice manifest macrocephaly, behavioral characteristics such as increased anxiety-like behavior and altered social behavior, as well as cognitive deficits, but the behavioral phenotypes of the various *Chd8* mutant mouse lines generated by different groups overlap only partially. We have now cited the studies mentioned by the reviewer and clarified this issue in the Introduction section of the revised manuscript (page 3, lines 40-43).

-Line 155: need to qualify that memory formation was only assessed in the Morris water maze test.

[Response] As suggested by the reviewer, we have now modified the text in the revised manuscript (page 9, lines 188-191).

-Line 164: should mention the specific behavioral abnormalities found.

[Response] In this study, we found that male *Chd8* CKO mice manifested increased anxiety-like behavior in the open-field test. We have mentioned this specific result in the revised manuscript (page 9, lines 199-203).

-Line 190: The studies reporting these findings should be properly cited. Two important studies from leading laboratories in the field that clearly showed these effects are not cited: Suetterlin et al., Cereb Cortex 2018 and Jimenez et al. Mol Autism, 2021.

[Response] We have now cited these references in the revised manuscript as suggested by the reviewer (page 10, lines 233-235).

Response to Reviewer #3

Kawamura and Nishiyama report on conditional Chd8 deletion in postmitotic neurons and adult mice in an effort to distinguish postnatal from prenatal consequences of Chd8 mutation. Their focus are activity-dependent processes for which they expose cells to elevated KCl concentrations to induce depolarization and investigate gene transcription and chromatin accessibility. In adult cKO mice, they investigate expression of activity-dependent genes and behavior. While the study is very well conducted and presented the findings remain moderate and mostly confirmatory of previous studies. Importantly, no overt deleterious effects of conditional Chd8 mutation in adult mice can be observed, possibly suggesting that disease relevant abnormalities maybe laid down during prenatal or early postnatal development.

[Response] We thank the reviewer for the careful review of our manuscript and for the statements that “the study is very well conducted and presented,” and that “Importantly, no overt deleterious effects of conditional Chd8 mutation in adult mice can be observed, possibly suggesting that disease relevant abnormalities maybe laid down during prenatal or early postnatal development.” We also thank the reviewer for suggestions that we feel have helped us to greatly improve our manuscript.

1. *I would recommend slightly modifying the diagram in Fig. 1a, by making it clear that this is not the process used in all experiments. Maybe TTX and KCl can be added in different color or font, or the diagram can be duplicated with these additions for Fig. 2.*

[Response] As suggested, we have now highlighted TTX/D-AP5 and KCl treatments in different colors and clarified the corresponding figures for the experiments performed under the 5 or 55 mM KCl conditions in the revised manuscript (**new Fig. 1a**).

Kawamura et al. Figure 1

2. *I would recommend in experiments using adult mice a breakdown according to sex and also in the cell culture experiments ideally including information on the sex of the embryos used.*

[Response] In the original manuscript, male and female mice at E18.5 were used for primary culture of neurons, and male mice for experiments *in vivo*. We have now added new behavioral data for female mice (**new Figures 5 and 6**). We have also now described the breakdown of data according to sex for each experiment in more detail in the Methods section and figure legends of the revised manuscript (page 16, lines 389-390; page 28, lines

637-638; page 30, lines 651-652; page 32, line 683-page 33, line 686; page 34, lines 700-703; page 37, lines 712-715).

3. Interestingly, *Chd8* mutant mice show decreased susceptibility to lethal seizures. I would appreciate if the authors could elaborate more on this point. Is this purely the consequence of downregulated expression of activity-dependent genes?

[Response] We apologize for any inadequacy in our description that might have led to some misunderstanding. As mentioned in the original manuscript, the proportion of animals who achieved seizure stage 6 (death) after KA treatment was reduced for *Chd8* CKO mice (1 of 12 mice) compared with control mice (6 of 14 mice). However, this difference in seizure severity did not reach statistical significance ($P = 0.2213$) (Fig. 4b, left). We also did not detect a significant difference in seizure severity between *Chd8* CKO and control female mice after KA treatment (**new Fig. 4b, right**). These results suggest that *Chd8* ablation in the adult brain may not affect seizure susceptibility. To avoid any misunderstanding, we have removed both the text mentioned above and the corresponding figure from the Results section, and we have clarified the data for seizure susceptibility in *Chd8* CKO mice in the revised manuscript (page 7, lines 141-142).

Kawamura *et al.* Figure 4

4. I would also appreciate a little more discussion of the identified dysregulated pathways. How well do the pathways in Fig. 1 and 2 align with theories of *Chd8*-mediated transcriptional control or ASD dysregulation.

[Response] CHD8 is thought to serve as a transcriptional activator, given that it preferentially interacts with promoter regions of actively transcribed genes. Consistent with previous findings that the expression of genes related to transcription, mRNA processing, and nervous system development was downregulated by CHD8 haploinsufficiency, our results have now shown downregulation of similar pathways in neurons of *Chd8* CKO mice. In addition to these expression changes, genes related to translation were highly enriched among downregulated genes in *Chd8* CKO neurons. Local translation in neurons supplies proteins to axons and synapses that are required for synaptic plasticity and neuronal function (Liu-Yesucevitz *et al.*, *J. Neurosci.* 31: 16086–16093, 2011). Dysregulation of translation has previously been implicated in ASD-like phenotypes (Gkogkas *et al.*, *Nature* 493: 371–377, 2013). CHD8 therefore appears to transcriptionally regulate multiple processes

including transcription, translation, and nervous system development, and effects of *CHD8* mutation on these processes might underlie ASD pathogenesis. We have now addressed this issue in the Discussion section of the revised manuscript (page 9, line 205-page 10, line 212).

Response to Reviewer #4

In this manuscript A. Kawamura & M. Nishiyama examine function of the chromatin modifier ASD-related gene Chd8 in postmitotic neurons using both in vitro and in vivo approaches that utilize the well-known Cre recombination system induced by tamoxifen. These authors have published relevant work in the Chd8 field, and this manuscript would nicely add to their Chd8-biology sequelae when appropriately revised. Although the downregulation of expression of neuronal genes as a consequence of Chd8 deletion is expected, and a finding that has been previously described in constitutive Chd8 ablation models; this study is novel in the discovery that links Chd8 deletion specifically in postmitotic neurons with the alteration of the expression of activity-dependent genes induced by KCl-mediated neuronal depolarization. The study is also novel in linking ablation of CHD8 in adult mice with attenuation of activity-dependent transcriptional responses in the hippocampus to kainic acid-induced seizures. Experiments are overall well-controlled and conducted, with some addressable exceptions mentioned below, and findings are mostly presented in a manner that is clear and organized. Overall, the study is somewhat strong but well designed and it should advance to the field of filling gaps in the understanding Chd8 biology. The results are potentially of broad interest to the fields of ASD preclinical models and more generally neurodevelopmental and psychiatric disorders. This reviewer would be happy to review a revised version of this manuscript.

[Response] We thank the reviewer for the careful review of our manuscript and for the statements that “the study is somewhat strong but well designed and it should advance to the field of filling gaps in the understanding Chd8 biology,” and that “The results are potentially of broad interest to the fields of ASD preclinical models and more generally neurodevelopmental and psychiatric disorders.” We also thank the reviewer for suggestions that we feel have helped us to greatly improve our manuscript.

Specific comments/issues, all addressable through adding minimal experimental work and/or acknowledging limitations/deficiencies of the study:

1. Please add rationale behind hippocampal in vitro cultures being made at embryonic day E18.5.

[Response] Primary cultures of mouse hippocampal neurons are generally prepared from embryos at the late developmental stage (E17.5–18.5) because the generation of pyramidal neurons is almost complete at this stage and embryonic tissue has advantages in that it is dissociated more readily and contamination with glial cells and fibroblasts is reduced. Primary neurons isolated from the hippocampus of mice at E18.5 have been also used for analysis of activity-dependent transcription in previous studies (Wu *et al.*, *Neuron* 56: 94–108, 2007). We have now addressed this point in the Methods section of the revised manuscript (page 12, lines 263-266).

2. For their in vitro studies, authors consistently refer to primary cultures as neurons culture. Please add any missing methodology or acknowledge through out the manuscript these cultures would also include other cell types (i.e., glia, astrocytes, oligodendrocytes, etc). This is important for assessing Chd8 function in a specific cell type, in this case, and as

claimed, in Neurons. Imaging of this neuronal cultures with specific cellular markers would help visualizing the cultures they are working with.

[Response] We performed immunostaining of the primary cultured cells and quantified neurons and glial cells. As suggested by the reviewer, the cultures included about ~17% astrocytes (GFAP-positive cells) and ~1% oligodendrocytes (Olig2-positive cells) in addition to neurons (TUBB3-positive cells) (**new Fig. 1c and Supplementary Fig. 1a, b**). The proportion of each cell type did not differ between *Chd8* CKO and control mice. We have now clarified this issue in the revised manuscript (page 4, lines 68-71).

Kawamura et al. Figure 1

Kawamura et al. Supplementary Figure 1

3. Please add to the rationale of using KA treatment. Is there a clinical based motivation? Do *Chd8* mutation carries present seizures?

[Response] Seizure induction by kainic acid is one of the most commonly studied models of temporal lobe epilepsy (Lévesque *et al.*, *Neurosci. Biobehav. Rev.* 37: 2887–2899, 2013). Whereas spontaneous seizures have not been previously shown in *Chd8* heterozygous mutant mice, some individuals with ASD associated with *CHD8* mutation experience seizures (Bernier *et al.*, *Cell* 158: 263–276, 2014). These observations motivated us to examine seizure susceptibility in *Chd8* CKO mice. We have now addressed these points in the Discussion and Methods sections of the revised manuscript (page 10, lines 231-233; page 16, lines 378-379).

4. Confirmation of changes at protein levels for the activity dependent genes RNA-Seq findings would validate such changes in expression after the KCl treatment (*Arc*, *Egr1*, *Fos*, *Fosb*, *Npas4*, and *Nr4a1*). Do these changes persist? A time course would be needed to answer this question.

[Response] In response to this comment, we examined the expression of such proteins in *Chd8* CKO and control neurons. Immunoblot analysis detected two immunoreactive isoforms of FOSB protein (full-length FOSB and a shorter form, FOSB2), both of which were induced after treatment of neurons with 55 mM KCl for 2 h. Consistent with the results of our RNA-seq analysis, the abundance of FOSB was significantly downregulated and that of FOSB2 tended to be reduced in *Chd8* CKO neurons compared with control neurons (**new Supplementary Fig. 3d–f**). We also examined the expression of EGR1 and NR4A1 by immunoblot analysis with two commercially available antibody preparations, but we did not detect any signals, likely as a result of low reactivity of the antibodies. Given that FOSB expression was significantly downregulated in *Chd8* CKO neurons, we believe that the expression of other activity-dependent genes at the protein level is also likely to be downregulated by *Chd8* ablation. In addition, we performed RT-qPCR analysis to examine the time course of the expression changes for activity-dependent genes in *Chd8* CKO and control neurons. The peak expression levels were apparent at ~1 h for *Egr1* mRNA and ~2 h for *Fosb* and *Nr4a1* mRNAs, after which the expression levels decreased gradually (**new Supplementary Fig. 3a–c**). We have now addressed these points in the revised manuscript (page 5, lines 90-91; page 5, lines 96-98).

Kawamura *et al.* Supplementary Figure 3

5. The claim in line 97: “These results thus suggested that CHD8 is required for the induction of activity-dependent gene transcription *in vitro*”, seems premature at the stage of the finding presented so far; it could be moved to discussion section and expand on the reasoning to conclude this.

[Response] As suggested by the reviewer, we have moved this sentence to the Discussion and toned down our conclusion. Our RNA-seq data revealed that the expression of activity-dependent genes was downregulated in *Chd8* CKO neurons (Fig. 2). CHD8 bound to the promoter regions of these genes (Fig. 3), suggesting that CHD8 may regulate the induction of activity-dependent gene transcription. We have now described these points in the Discussion section of the revised manuscript (page 10, lines 226-229).

6. Please expand in the discussion section of the *atac-Seq* negative findings – what are the limitations of this design?

[Response] Consistent with our observations, previous ATAC-seq analysis revealed only small changes in chromatin accessibility in cultured neurons exposed to an elevated extracellular concentration of KCl (ref. 28). This might be due to a difference in the chromatin accessibility state of cultured neurons or to technical problems. As an alternative approach to examine whether *Chd8* ablation alters chromatin accessibility, we performed ATAC-seq analysis for the hippocampus of *Chd8* CKO and control mice with a similar seizure stage at 60 min after KA treatment. Chromatin accessibility at the gene body of several activity-dependent genes including *Fosb*, *Nr4a1*, and *Egr1* was increased by KA treatment (**new Fig. 4d**). We also found that ATAC-seq signal intensity at these loci was decreased in the hippocampus of *Chd8* CKO mice compared with that of control mice. These results thus suggested that changes in chromatin accessibility due to *Chd8* ablation might affect the induction of activity-dependent genes *in vivo*. Although we did not detect changes in chromatin accessibility in response to *Chd8* deletion *in vitro*, likely as a result of technical limitations, it is possible that CHD8 also affects chromatin accessibility *in vitro* to some extent, a possibility that warrants further investigation. We have now addressed these points in the Results and Discussion section of the revised manuscript (page 7, lines 147-155; page 10, lines 218-226).

Kawamura *et al.* Figure 4

7. Please add the rationale of administering tamoxifen for 5 days to 8–11-week-old mice. Explain why this age and why the 5 days treatment.

[Response] CHD8 has previously been shown to regulate the proliferation and differentiation of neural progenitor cells as well as neuron development at embryonic or early postnatal stages. In contrast, the function of CHD8 in the adult brain has remained unclear. To examine the role of CHD8 at the adult stage, when brain development and circuit formation are essentially complete, we administered tamoxifen to 8- to 11-week-old *CAG-CreER/Chd8^{F/F}* mice. Furthermore, we administered tamoxifen for five consecutive days because the deletion of floxed alleles by tamoxifen is more efficient after multiple injections than after a single treatment. This protocol of tamoxifen administration has often been used in previous studies to delete a gene of interest in adult mice (Jahn *et al.*, *Sci. Rep.* 8: 5913, 2018). We have now clarified these points in the revised manuscript (page 11, lines 255-256).

8. Figure 4a – please add the rationale behind assessing tamoxifen induced *Chd8* deletion at the mRNA level only – what is the half-life of the protein? If unknown, then this experiment should include *Chd8* protein expression assessed via immunocytochemistry or Western Blot.

[Response] As suggested by the reviewer, we examined CHD8 expression at the protein level by immunoblot analysis. The abundance of CHD8 protein was reduced by ~80% in the *Chd8* CKO mouse hippocampus (**new Supplementary Fig. 6a, b**), indicating that CHD8 protein was efficiently depleted by tamoxifen administration. We have now addressed this issue in the Results section of the revised manuscript (page 7, lines 136-138).

Kawamura *et al.* Supplementary Figure 6

9. From the disease model perspective, do the heterozygous conditional tamoxifen treated animals exhibit any phenotypes. Were these assessed/tested? Please clarify and explain if heterozygous mice were not analyzed.

[Response] Although the phenotypes of inducible *Chd8* heterozygous knockout mice are of interest, we believe that they are beyond the scope of the present manuscript. We have already begun a behavioral analysis of *CAG-CreER/Chd8^{+/F}* mice treated with tamoxifen at several developmental time points, but these data are still preliminary and need further validation. We would like to include these results in our next paper.

10. “Although we did not detect a significant difference in seizure severity between *Chd8* CKO and control mice after KA treatment, the proportion of animals who achieved seizure stage 6 (death) was reduced for *Chd8* CKO mice (1 of 12 mice) compared with control mice (6 of 14 mice) (Fig.4b).” this should be deeply discussed in the discussion section.

[Response] We apologize for an inadequate description that might have led to some misunderstanding. As mentioned in the original manuscript, the proportion of animals who achieved seizure stage 6 (death) after KA treatment was reduced for *Chd8* CKO mice (1 of 12 mice) compared with control mice (6 of 14 mice). However, this difference did not reach statistical significance ($P = 0.2213$) (Fig. 4b, left). We also did not detect a significant difference in seizure severity after KA treatment between *Chd8* CKO and control female mice (**new Fig. 4b, right**). These results suggested that *Chd8* ablation in the adult brain may not affect seizure susceptibility. To avoid misunderstanding, we have omitted the above sentence and the corresponding figure and have clarified this issue in the revised manuscript (page 7, lines 141-142).

Kawamura *et al.* Figure 4

11. For drawing functional conclusions, *in vivo* mRNA expression changes (*Fosb*, *Nr4a1*, and *Egr1*) after tamoxifen and KCL/KA treatment should also be confirmed at protein levels. Otherwise acknowledge and explain the reasoning fundamentally.

[Response] As suggested by the reviewer, we examined the expression of these proteins in the hippocampus of *Chd8* CKO mice. Immunoblot analysis detected two immunoreactive isoforms of FOSB protein (full-length FOSB and a shorter form, FOSB2), as was observed in cultured neurons. The abundance of both isoforms of FOSB was significantly reduced in the hippocampus of *Chd8* CKO mice compared with that of control mice (**new Supplementary Fig. 6a–c**). These results are consistent with those obtained at the mRNA level by RT-qPCR analysis (Fig. 4c). We also examined the expression of EGR1 and NR4A1 by immunoblot analysis with two commercially available antibody preparations, but we did not detect any signals, likely as a result of low reactivity of the antibodies. Given that FOSB expression was significantly downregulated in the hippocampus of *Chd8* CKO mice, we believe that the expression of other activity-dependent genes at the protein level would also be downregulated by *Chd8* ablation. We have now addressed these points in the revised manuscript (page 7, lines 145-147).

Kawamura et al. Supplementary Figure 6

12. Importantly, this manuscript does not consider sex as a biological variable in their in vivo analyses. All the presented data from males. What is the reasoning for this? This should be stated by the authors in the manuscript acknowledging current field-reporting guidelines, where sex should be considered as a biological variable regardless ASD being more prevalent in males (something that is also changing).

[Response] Given that the prevalence of ASD is strongly male-biased, we mainly used male mice in the original manuscript. In response to this comment, we have now also performed behavioral tests with *Chd8* CKO and control female mice as well as males. We did not detect a significant difference in seizure severity between *Chd8* CKO and control female mice after KA treatment (**new Fig. 4b**). In the Morris water-maze test, the visible and hidden platform trials over five consecutive days revealed that escape latency was similar in *Chd8* CKO and control female mice (**new Fig. 5a**). The number of target platform crossings during a probe trial also did not differ between *Chd8* CKO and control female mice (**new Fig. 5b**). Moreover, the time spent in the target quadrant during the probe trial was significantly increased relative to that in each of the other quadrants for both *Chd8* CKO and control female mice (**new Fig. 5c**), suggestive of normal learning and memory formation in *Chd8* CKO mice. Total distance traveled in the open-field test did not differ between genotypes, suggestive of normal locomotor activity in *Chd8* CKO male and female mice (**new Fig. 6a**). Whereas male *Chd8* CKO mice showed a decrease in the time spent in the center of the open field compared with control mice, female *Chd8* CKO mice showed no such difference relative to control mice (**new Fig. 6b**). We did not detect a difference between genotypes of either sex for time spent in the open arms during the elevated plus-maze test or for that spent in the light room during the light-dark transition test (**new Fig. 6c, d**). Male and female *Chd8* CKO mice showed normal self-grooming behavior (**new Fig. 6e**). Female, but not male, *Chd8* CKO mice showed an increase in the total contact time during the reciprocal social-interaction test (**new Fig. 6f**), whereas the number of social contacts did not differ between *Chd8* CKO and control mice of either sex (**new Fig. 6g**). In the three-chamber sociability test, both *Chd8* CKO and control animals of each sex showed a significant preference for a novel mouse (stranger 1) (**new Fig. 6h, i**). In the social-novelty preference test, both *Chd8* CKO and control male mice also showed a significant preference for a novel mouse (stranger 2) over a familiar mouse (stranger 1), whereas female mice of both genotypes did not show a significant preference for a novel mouse (stranger 2) (**new Fig. 6j, k**). These results thus suggested that *Chd8* ablation in the adult brain has selective effects on behavioral characteristics. We have now addressed these points in the revised

Kawamura *et al.* Figure 4

Kawamura *et al.* Figure 5

Open-field test

Elevated plus-maze test

Light-dark transition test

Self-grooming

Social-interaction test

Three-chamber test (sociability)

Three-chamber test (social-novelty preference)

Kawamura et al. Figure 6

13. Last paragraph of the discussion (and overall sense throughout the manuscript):
“Collectively, the present findings reveal a role for CHD8 in activity-dependent transcriptional regulation in postmitotic neurons, and they therefore provide potentially important insight into the molecular mechanisms underlying the pathogenesis of ASD”. Please discuss on how the behavioral changes observed in a tamoxifen induced *Chd8* deletion adult mouse would be relevant to ASD modeling from the point of view of ASD being a developmental disorder. If this aspect of their research is something the authors do not consider relevant to the disease modeling, then the overall angle of the manuscript should change and discussed accordingly in the discussion section.

[Response] CHD8 was previously shown to have dosage-dependent effects on transcriptional regulation and behavioral phenotypes (Sood *et al.*, *Proc. Natl. Acad. Sci. USA* 117: 22331–22340, 2020; Hurley *et al.*, *Mol. Autism* 12: 16, 2021). Given that homozygous deletion of *Chd8* in the adult brain resulted in transcriptional changes and some behavioral deficits, we believe that *Chd8* heterozygous mutation might have similar but less pronounced effects, a possibility that warrants further investigation. Our findings thus provide potentially important insight into the molecular mechanisms and behavioral phenotypes related to the pathogenesis of ASD. We have now addressed this issue in the Discussion section of the revised manuscript (page 11, lines 242-245).

Reviewers' comments:

Reviewer #1 (Remarks to the Author):

The authors have fully addressed my review comments. I do not have any additional comments.

Reviewer #3 (Remarks to the Author):

The authors addressed this reviewer's suggestions and sufficiently revised their manuscript.

Reviewer #4 (Remarks to the Author):

Authors have adequately addressed all concerns from this reviewer, and I consider the new version suitable for publication.

Reviewer #5 (Remarks to the Author):

First, I would like to say that this manuscript is strong overall and will be a nice contribution to the field. In general, the experiments address a question of interest using multiple systems (conditional ablation in vivo and in vitro, activity-dependent vs. baseline) and across multiple methods (genomics, behavior). There are a few points I think still should be addressed.

1. There is no information about replicates for the ATAC-seq experiments and there is no genome-wide data shown for the in vivo findings. To evaluate the strength and reproducibility of differences, replicates must be reported/incorporated and results should not only show hand-picked candidates, but instead capture systemic trends. This is my only point that requires more than text edits, but I think is critical if the ATAC-seq data (both in vitro and in vivo) is to be included as evidence in either direction (for or against accessibility changes in the conditional mutants).
2. The abstract should make clear that the studies described in this work only include comparisons of homozygous conditional deletion to WT neurons/mice. This may be implied, but I think should be explicit.
3. The model is described as "post-mitotic neuron" conditional ablation. While the authors provide evidence from the culture model that the majority of cells express neuronal markers, greater care should be taken in ascribing phenotypes from the in vivo tamoxifen-induced conditional model. As this would drive ablation in all cells, rather than neuronal specificity, Interpretation of the mouse studies needs to be updated in address to fact that this model is a temporally-defined conditional, rather than cell-type specific. Thus, phenotypes described could be driven by neuronal, non-neuronal, and even non-CNS cell populations. Such differences could also drive observed differences in the ATAC-seq data.
4. Reporting the ChIP-qPCR results as "not differing" seems to rely on use of $P < 0.05$ significance threshold and gives a false sense of a clear negative finding. Given the apparent sample size of three, this experiment seems underpowered to detect anything but a very large effect. Indeed, some of the panels show relatively consistent differences, albeit ones that don't quite hit the arbitrary $P < 0.05$ threshold. I think the results are more ambiguous than presented. One solution would be to move to main figure and to mention some loci trended towards different enrichment, but not at a $P < 0.05$ threshold.

Response to Reviewer #1

The authors have fully addressed my review comments. I do not have any additional comments.

[Response] We thank the reviewer for the positive evaluation of our manuscript. We also appreciate the time and effort taken in reviewing our paper.

Response to Reviewer #3

The authors addressed this reviewer's suggestions and sufficiently revised their manuscript.

[Response] We thank the reviewer for the positive evaluation of our manuscript. We also appreciate the time and effort taken in reviewing our paper.

Response to Reviewer #4

Authors have adequately addressed all concerns from this reviewer, and I consider the new version suitable for publication.

[Response] We thank the reviewer for the positive evaluation of our manuscript. We also appreciate the time and effort taken in reviewing our paper.

Response to Reviewer #5

First, I would like to say that this manuscript is strong overall and will be a nice contribution to the field. In general, the experiments address a question of interest using multiple systems (conditional ablation in vivo and in vitro, activity-dependent vs. baseline) and across multiple methods (genomics, behavior). There are a few points I think still should be addressed.

[Response] We thank the reviewer for the careful review of our manuscript and for the statement that “This manuscript is strong overall and will be a nice contribution to the field.” We also thank the reviewer for suggestions that we feel have helped us to greatly improve our manuscript.

1. There is no information about replicates for the ATAC-seq experiments and there is no genome-wide data shown for the in vivo findings. To evaluate the strength and reproducibility of differences, replicates must be reported/incorporated and results should not only show hand-picked candidates, but instead capture systemic trends. This is my only point that requires more than text edits, but I think is critical if the ATAC-seq data (both in vitro and in vivo) is to be included as evidence in either direction (for or against accessibility changes in the conditional mutants).

[Response] In the original manuscript, ATAC-seq analysis was performed with two independent replicates for experiments *in vitro* but with one sample for each condition for experiments *in vivo*. During revision of the manuscript, we performed additional ATAC-seq analysis with one more sample each for the hippocampus of *Chd8* CKO and control mice with a similar seizure stage at 60 min after KA or vehicle treatment. We have now clarified the information regarding replicates for the ATAC-seq experiments in the figure legends of the revised manuscript. Furthermore, we now show the results for the two independent ATAC-seq replicates (**new Supplementary Figs. 5 and 7**) as well as the merged data for these replicates (Fig. 3a–c and **new Fig. 4d–f**) for each condition. Heat map and density profiles around high-confidence CHD8 ChIP-seq peaks revealed a small decrease in ATAC-seq signal intensity in the hippocampus of *Chd8* CKO mice compared with that of control mice (**new Fig. 4e**). These results were consistently observed in the two independent replicates (**new Supplementary Fig. 7a**) and suggested that *Chd8* ablation attenuates genome-wide chromatin accessibility *in vivo*. We have now addressed these points in the revised manuscript (page 7, line 151–page 8, line 162).

Kawamura *et al.* Figure 4

Kawamura *et al.* Supplementary Figure 7

2. The abstract should make clear that the studies described in this work only include comparisons of homozygous conditional deletion to WT neurons/mice. This may be implied, but I think should be explicit.

[Response] As suggested by the reviewer, we have now clarified in the Abstract of the revised manuscript that the data in this study are for homozygous deletion of *Chd8* (page 2, lines 19-24).

3. The model is described as "post-mitotic neuron" conditional ablation. While the authors provide evidence from the culture model that the majority of cells express neuronal markers, greater care should be taken in ascribing phenotypes from the *in vivo* tamoxifen-induced conditional model. As this would drive ablation in all cells, rather than neuronal specificity,

Interpretation of the mouse studies needs to be updated in address to fact that this model is a temporally-defined conditional, rather than cell-type specific. Thus, phenotypes described could be driven by neuronal, non-neuronal, and even non-CNS cell populations. Such differences could also drive observed differences in the ATAC-seq data.

[Response] We tried to use the terms “postmitotic neurons” for the experiments *in vitro* and “adult brain” for those *in vivo* in the original manuscript. As mentioned by the reviewer, the *CAG-CreER* mouse line drives recombination in all cell types, rather than showing cell type specificity. It is therefore possible that *Chd8* deletion not only in postmitotic neurons but in glial cells and other cell populations might influence transcriptomic, epigenetic, and behavioral changes observed in our study. Further studies are warranted to determine whether these changes are indeed attributable to alterations in postmitotic neurons. We have now clarified these points in the Discussion section of the revised manuscript (page 11, lines 254-259).

4. Reporting the ChIP-qPCR results as “not differing” seems to rely on use of $P < 0.05$ significance threshold and gives a false sense of a clear negative finding. Given the apparent sample size of three, this experiment seems underpowered to detect anything but a very large effect. Indeed, some of the panels show relatively consistent differences, albeit ones that don’t quite hit the arbitrary $P < 0.05$ threshold. I think the results are more ambiguous than presented. One solution would be to move to main figure and to mention some loci trended towards different enrichment, but not at a $P < 0.05$ threshold.

[Response] We have now moved the ChIP-qPCR results to a main figure (Fig. 3d, e) and describe the trend for differences in their enrichment in the Results section of the revised manuscript (page 6, line 130-page 7, line 133).

REVIEWERS' COMMENTS:

Reviewer #5 (Remarks to the Author):

I am satisfied with the edits in the revised version.